# Instant Complexity Reduction in CNNs using Locality-Sensitive Hashing

## Abstract

To reduce the computational cost of convolutional neural networks (CNNs) for usage on resource-constrained devices, structured pruning approaches have shown promising results, drastically reducing floating-point operations (FLOPs) without substantial drops in accuracy. However, most recent methods require fine-tuning or specific training procedures to achieve a reasonable trade-off between retained accuracy and reduction in FLOPs. This introduces additional cost in the form of computational overhead and requires training data to be available. To this end, we propose HASTE (**Has**hing for **T**ractable **E**fficiency), a parameter-free and data-free module that acts as a plug-and-play replacement for any regular convolution module. It instantly reduces the network's test-time inference cost without requiring any training or fine-tuning. We are able to drastically compress latent feature maps without sacrificing much accuracy by using locality-sensitive hashing (LSH) to detect redundancies in the channel dimension. Similar channels are aggregated to reduce the input and filter depth simultaneously, allowing for cheaper convolutions. We demonstrate our approach on the popular vision benchmarks CIFAR-10 and ImageNet. In particular, we are able to instantly drop 46.72% of FLOPs while only losing 1.25% accuracy by just swapping the convolution modules in a ResNet34 on CIFAR-10 for our HASTE module.

## 1 Introduction

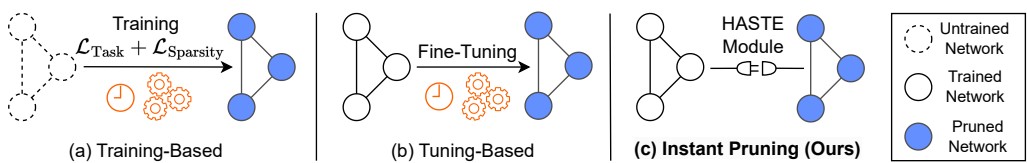

Figure 1: Overview of related pruning approaches. Training-based methods require specialized training procedures. Methods based on fine-tuning need retraining to compensate lost accuracy in the pruning step. Our method instantly reduces network FLOPs and maintains high accuracy entirely without training or fine-tuning.

With the rise in availability and capability of deep learning hardware, the possibility to train ever larger models led to impressive achievements in the field of computer vision. At the same time, concerns regarding high computational costs, environmental impact and the applicability on resource-constrained devices are growing. This led to the introduction of carefully constructed efficient models (Howard et al., 2017; Sandler et al., 2018; Tan & Le, 2019; 2021; Zhang et al., 2018; Ma et al., 2018) that offer fast inference in embedded applications, gaining speed by introducing larger inductive biases. Yet, highly scalable and straight-forward architectures (Simonyan & Zisserman, 2015; He et al., 2016; Dosovitskiy et al., 2021; Liu et al., 2021a; 2022; Woo et al., 2023) remain popular due to their performance and ability to generalize, despite requiring more data, time and energy to train. To still allow for larger models to be used in mobile applications, various methods (Zhang et al., 2016; Lin et al., 2017b; Pleiss et al., 2017; Han et al., 2020; Luo et al., 2017) have been proposed to reduce their computational cost. One particularly promising field of research for the

compression of convolutional architectures is pruning (Wimmer et al., 2023), especially in the form of structured pruning for direct resource savings (Anwar et al., 2017).

However, the application of existing work is restricted by two factors. Firstly, many proposed approaches rely on actively learning which channels to prune during the regular model training procedure (Dong et al., 2017; Liu et al., 2017; Gao et al., 2019; Verelst & Tuytelaars, 2020; Bejnordi et al., 2020; Li et al., 2021; Xu et al., 2021). This introduces additional parameters to the model, increases the complexity of the optimization process due to supplementary loss terms, and requires existing models to be retrained to achieve any reduction in FLOPs. The second limiting factor is the necessity of performing fine-tuning steps to restore the performance of pruned models back to acceptable levels (Wen et al., 2016; Li et al., 2017; Lin et al., 2017a; Zhuang et al., 2018; He et al., 2018). Aside from the incurred additional cost and time requirements, this creates a dependency on the availability of the data that was originally used for training the baseline model, as tuning the model on a different set of data can lead to catastrophic forgetting (Goodfellow et al., 2014).

To this end, we propose HASTE, a plug-and-play channel pruning approach that is entirely data-free and does not require any real or synthetic training data. Our method instantly reduces the computational complexity of convolution modules without requiring any additional training or fine-tuning. To achieve this, we utilize a locality-sensitive hashing scheme (Indyk & Motwani, 1998) to dynamically detect and cluster similarities in the channel dimension of latent feature maps in CNNs. By exploiting the distributive property of the convolution operation, we take the average of all input channels that are found to be approximately similar and convolve it with the sum of corresponding filter channels. This reduced convolution is performed on a smaller channel dimension, which drastically lowers the amount of FLOPs required.

Our experiments demonstrate that the HASTE module is capable of greatly reducing computational cost of a wide variety of pre-trained CNNs while maintaining high accuracy. More importantly, it does so directly after exchanging the original convolutional modules for the HASTE block. This allows us to skip lengthy model trainings with additional regularization and sparsity losses as well as extensive fine-tuning procedures. Furthermore, we are not tied to the availability of the dataset on which the given model was originally trained. Our pruning approach is entirely data-free, thus enabling pruning in a setup where access to the trained model is possible, but access to the data is restricted. To the best of our knowledge, this makes the HASTE module the first dynamic and data-free CNN pruning approach that does not require any form of training or fine-tuning.

Our main contributions are:

- We propose a locality-sensitive hashing based method to detect redundancies in the latent features of current CNN architectures. Our method incurs a low computational overhead and is entirely data-free.

- We propose HASTE, a scalable, plug-and-play convolution module replacement that leverages these structural redundancies to save computational complexity in the form of FLOPs at test time, without requiring any training steps.

- We demonstrate the performance of our method on popular CNN models trained on benchmark vision datasets. We also identify a positive scaling behavior, achieving higher cost reductions on deeper and wider models.

## 2 RELATED WORK

When structurally pruning a model, its computational complexity is reduced at the expense of performance on a given task. For this reason, fine-tuning is often performed after the pruning scheme was applied. The model is trained again in its pruned state to compensate the lost structural components, often requiring multiple epochs of tuning (Li et al., 2017; Zhuang et al., 2018; Xu et al., 2021) on the dataset originally used for training. These methods tend to remove structures from models in a static way, not adjusting for different degrees of sparsity across varying input data. Some recent methods try to avoid fine-tuning by learning a pruning pattern during regular model training (Liu et al., 2017; Gao et al., 2019; Xu et al., 2021; Li et al., 2021; Elkerdawy et al., 2022). This generates an input-dependent dynamic path through the network, allocating less compute to sparser images.

**Static Pruning.** By finding general criteria for the importance of individual channels, some recent methods propose static pruning approaches. PFEC (Li et al., 2017) prunes filter kernels with low importance measured by their $L^1$-norm in a one-shot manner. DCP (Zhuang et al., 2018) equips models with multiple loss terms before fine-tuning to promote highly discriminative channels to be formed. Then, a channel selection algorithm picks the most informative ones. FPGM (He et al., 2019) demonstrates a fine-tuning-free pruning scheme, exploiting norm-based redundancies to train models with reduced complexity. He et al. (2018) explore a compression policy generated by reinforcement learning. A handful of data-free approaches exist, yet they either use synthetic data to re-train the model (Bai et al., 2023) or generate a static model (Yvinec et al., 2023; Bai et al., 2023) that is unable to adapt its compression to the availability of hardware resources on the fly.

**Dynamic Gating.** To accommodate inputs of varying complexity in the pruning process, recent works try to learn dynamic, input-dependent paths through the network (Xu et al., 2021; Li et al., 2021; Elkerdawy et al., 2022; Liu et al., 2017; Hua et al., 2019; Verelst & Tuytelaars, 2020; Bejnordi et al., 2020; Liu et al., 2019). These methods learn (binary) masks that toggle structural components of the underlying CNN at runtime. DGNet (Li et al., 2021) equips the base model with additional spatial and channel gating modules based on average pooling that are trained end-to-end together with the model using additional regularization losses. Similarly, DMCP (Xu et al., 2021) learns mask vectors using a pruning loss and does not need fine-tuning procedures after training. FTWT (Elkerdawy et al., 2022) decouples the task and regularization losses introduced by previous approaches, reducing the complexity of the pruning scheme. While these methods do not require fine-tuning, they introduce additional complexity through pruning losses and the need for custom gating modules during training to realize FLOP savings. We instead focus on hashing to speed up inference at test time, with no training and data requirement at all.

**Hashing for Fast Inference.** In recent years, the usage of locality-sensitive hashing Indyk & Motwani (1998) schemes as a means to speed up model inference has gained some popularity. In Reformer, (Kitaev et al., 2020) use LSH to reduce the computational complexity of multi-head attention modules in transformer models by finding similar queries and keys before computing their matrix product. Müller et al. (2022) employ a multiresolution hash encoding to construct an efficient feature embedding for neural radiance fields (NeRFs), leading to orders of magnitude speedup compared to previous methods. Chen et al. (2020; 2021) use an LSH scheme to store activation patterns of a high-dimensional feedforward network, only computing the strongest activating neurons during the forward pass. Approaches related to LSH have also been explored for model compression. Liu et al. (2021b) employ a count sketch-type algorithm to approximate the forward pass of MLP networks by hashing the model's input vector. Liu et al. (2021c) extend on FPGM (He et al., 2019) and explore the use of $k$-means clustering for finding redundant input channels. However, this approach is limited to fixed pruning ratios, and does not allow for input-dependent compression.

## 3 METHOD

In this section, we present HASTE, a novel convolution module based on locality-sensitive hashing that acts as a plug-and-play replacement for any regular convolution module, instantly reducing the FLOPs during inference. Firstly, we give a formal definition of the underlying LSH scheme. Secondly, we illustrate how hashing is used to identify redundancies inside latent features of convolutional network architectures. Lastly, we present the integration of the hashing process into our proposed HASTE module, which allows us to compress latent features for cheaper computations.

### 3.1 LOCALITY-SENSITIVE HASHING VIA SPARSE RANDOM PROJECTIONS

Locality-sensitive hashing is a popular approach for approximate fast nearest neighbor search in high-dimensional spaces. A hash function $h : \mathbb{R}^d \to \mathbb{N}$ is locality-sensitive, if similar vectors in the input domain $\boldsymbol{x}, \boldsymbol{y} \in \mathbb{R}^d$ receive the same hash codes $h(\boldsymbol{x}) = h(\boldsymbol{y})$ with high probability. This is in contrast to regular hashing schemes which try to reduce hash collisions to a minimum, widely scattering the input data across their hash buckets. More formally, we require a measure of similarity on the input space and an adequate hash function $h$. A particularly suitable measure for use in convolutional architectures is the cosine similarity, as convolving the (approximately) normalized kernel with the normalized input is equivalent to computing their cosine similarity. Pairwise similarities between vectors are preserved through hashing by the allocation of similar hash codes.

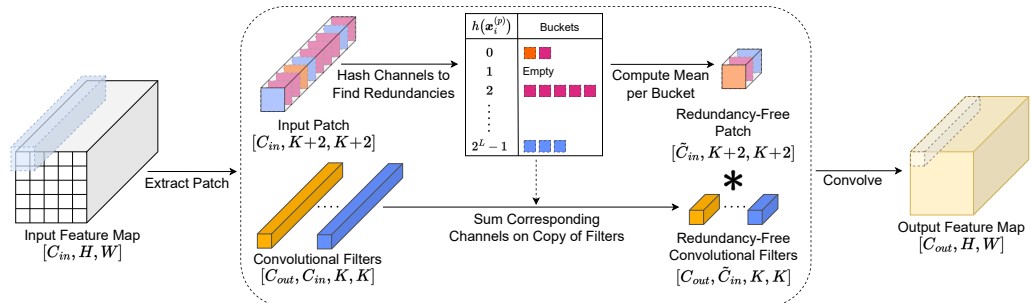

Figure 2: Overview of our proposed HASTE module. Each patch of the input feature map is processed to find redundant channels. Detected redundancies are then merged together, dynamically reducing the depth of each patch and the convolutional filters.

One particular family of hash functions that groups input data by cosine similarity is given by random projections (RP). These functions partition the high-dimensional input space through $L$ random hyperplanes, such that each input vector is assigned to exactly one section of this partitioning, called a hash bucket. Determining the position of an input $\boldsymbol{x} \in \mathbb{R}^d$ relative to all $L$ hyperplanes is done by computing the dot product with their normal vectors $\mathbf{v}_l \in \mathbb{R}^d$, $l \in \{1, 2, \ldots, L\}$, whose entries are drawn from a standard normal distribution $\mathcal{N}(0, 1)$. By defining

$$h_l : \mathbb{R}^d \to \{0, 1\}, \ h_l(\boldsymbol{x}) := \begin{cases} 1, \text{ if } \mathbf{v}_l \cdot \boldsymbol{x} > 0, \\ 0, \text{ else,} \end{cases} \tag{1}$$

we get a binary information representing to which side of the $l$-th hyperplane input $\boldsymbol{x}$ lies. The hyperparameter $L$ governs the discriminative power of this method, dividing the input space $\mathbb{R}^d$ into a total of $2^L$ distinct regions, or hash buckets. By concatenating all individual functions $h_l$, we receive the RP hash function

$$h : \mathbb{R}^d \to \{0, 1\}^L, \ h(\boldsymbol{x}) = (h_1(\boldsymbol{x}), h_2(\boldsymbol{x}), \ldots, h_L(\boldsymbol{x})) . \tag{2}$$

Note that $h(\boldsymbol{x})$ is an $L$-bit binary code, acting as an identifier of exactly one of the $2^L$ hash buckets. Equivalently, we can transform this code into an integer, labeling the hash buckets from 0 to $2^L - 1$:

$$h : \mathbb{R}^d \to \{0, 1, \ldots, 2^L - 1\}, \ h(\boldsymbol{x}) = 2^{L-1} h_L(\boldsymbol{x}) + 2^{L-2} h_{L-1}(\boldsymbol{x}) + \cdots + 2^0 h_1(\boldsymbol{x}). \tag{3}$$

While LSH already reduces computational complexity drastically compared to exact nearest neighbor search, the binary code generation still requires $L \cdot d$ multiplications and $L \cdot (d-1)$ additions per input. To further decrease the cost of this operation, we employ the method presented by Achlioptas (2003); Li et al. (2006): Instead of using standard normally distributed vectors $\mathbf{v}_l$, we use very sparse vectors $\boldsymbol{v}_l$, containing only elements from the set $\{1, 0, -1\}$. Given a targeted degree of sparsity $s \in (0, 1)$, the hyperplane normal vectors $\boldsymbol{v}_l$ are constructed randomly such that the expected ratio of zero entries is $s$. The remaining $1 - s$ of vector components are randomly filled with either 1 or $-1$, both chosen with equal probability. This reduces the dot product computation to a total of $L \cdot (d(1-s) - 1)$ additions and 0 multiplications, as we only need to sum entries of $\boldsymbol{x}$ where $\boldsymbol{v}_l$ is non-zero with the corresponding signs. Consequently, this allows us to trade expensive multiplication operations for cheap additions.

## 3.2 FINDING REDUNDANCIES WITH LSH

After establishing LSH via sparse random projections as a computationally cheap way to find approximate nearest neighbors in high-dimensional spaces, we now aim to leverage this method as a means of finding redundancies in the channel dimension of latent feature maps in CNNs. Formally, a convolutional layer can be described by sliding multiple learned filters $F_j \in \mathbb{R}^{C_{in} \times K \times K}$, $j \in \{1, 2, \ldots, C_{out}\}$ over the (padded) input feature map $X \in \mathbb{R}^{C_{in} \times H \times W}$ and computing the discrete convolution at every point. Here, $K$ is the kernel size, $H$ and $W$ denote the spatial dimensions of the input, and $C_{in}, C_{out}$ describe the input and output channel dimensions, respectively.

For any filter position, the corresponding input window contains redundant information in the form of similar channels. However, a regular convolution module ignores potential savings from reducing the amount of similar computations in the convolution process. We challenge this design choice and instead leverage redundant channels to save computations in the convolution operation. As the first step, we rasterize the (padded) input image into patches $X^{(p)} \in \mathbb{R}^{C_{in} \times (K+2) \times (K+2)}$ with an overlap of two pixels on each side. This is equivalent to splitting the spatial dimension into patches of size $K \times K$, but keeping the filter overlap to its neighbors. The special case of $K = 1$ is discussed in Appendix B.2.

To group similar channels together, we flatten all individual channels $X_i^{(p)}$ into vectors of dimension $(K+2)^2$ and center them by the mean along the channel dimension. We denote the resulting vectors as $\boldsymbol{x}_i^{(p)}$. Finally, they are hashed using $h$, giving us a total of $C_{in}$ hash codes. We then check which hash code appears more than once, as all elements that appear in the same hash bucket are determined to be approximately similar by the LSH scheme. Consequently, grouping the vector representations of $X_i^{(p)}$ by their hash code, we receive sets of redundant feature map channels.

In particular, note that our RP LSH approach is invariant to the scaling of a given input vector. This means that input channels of the same spatial structure, but with different activation intensities, still land in the same hash bucket, effectively finding even more redundancies in the channel dimension.

### 3.3 THE HASTE MODULE

Our approach is motivated by the distributivity of the convolution operation. Instead of convolving various filter kernels with nearly similar input channels and summing the result, we can approximate this operation by computing the sum of kernels first and convolving it with the mean of these redundant channels. The grouping of input channels $X_i^{(p)}$ into hash buckets provides a straight-forward way to utilize this distributive property for the reduction of required floating-point operations when performing convolutions.

To avoid repeated computations on nearly similar channels, we dynamically reduce the size of each input context window $X^{(p)}$ by compressing channels found in the same hash bucket, as shown in Figure 2. The merging operation is performed by taking the mean of all channels in one bucket. As a result, the number of remaining input channels of a given patch is reduced to $\tilde{C}_{in} < C_{in}$. In a similar manner to the reduction of the input feature map depth, we add the corresponding channels of all convolutional filters $F_j$. Note that this does not require hashing of the filter channels, as we can simply aggregate those kernels that correspond to the collapsed input channels. This step is done on the fly for every patch $p$, retaining the original filter weights for the next patch.

The choice of different merging operations for input and filter channels is directly attributable to the distributive property, as the convolution between the average input and summed filter set retains a similar output intensity to the original convolution. When choosing to either average or sum both inputs and filters, we would systematically under- or overestimate the original output, respectively.

Finally, the reduced input patch is convolved with the reduced set of filters in a sliding window manner to compute the output. This can be formalized as follows:

$$\sum_{i=1}^{C_{in}} F_{j,i} * X_i^{(p)} \approx \sum_{l=0}^{2^L-1} \left( \left( \sum_{i \in S_l^{(p)}} F_{j,i} \right) * \left( \frac{1}{|S_l^{(p)}|} \sum_{i \in S_l^{(p)}} X_i^{(p)} \right) \right), \tag{4}$$

where $S_l^{(p)} = \{i \in \{1, 2, \ldots, C_{in}\} \mid h(\boldsymbol{x}_i^{(p)}) = l\}$ contains all channel indices that appear in the $l$-th hash bucket. We assume the scaling factor $1/|S_l^{(p)}|$ to be zero if the bucket is empty for simplicity of notation. Since we do not remove entire filters, but rather reduce their depth, the output feature map retains the same spatial dimension and number of channels as with a regular convolution module. The entire procedure is summarized in Algorithm 1.

This reduction of input and filter depth lets us define a compression ratio $r = 1 - (\tilde{C}_{in}/C_{in}) \in (0, 1)$, determining the relative reduction in channel depth. Note that this ratio is dependent on the amount of redundancies in the input feature map $X$ at patch position $p$. Our dynamic pruning of channels allows for different compression ratios across images and even in different regions of the same input.

Although the hashing of input channels and both merging operations create additional computational cost, the overall savings on computing the convolution operations with reduced channel dimension outweigh the added overhead. The main additional cost lies in the merging of filter channels, as this process is repeated $C_{out}$ times for every patch $p$. However, since this step is performed by computationally cheap additions, it lends itself to hardware-friendly implementations. For a detailed overview of the computational cost of our HASTE module, we refer to Appendix B.1.

---

**Algorithm 1** Pseudocode overview of the HASTE module.

---

**Input**: Feature map $X \in \mathbb{R}^{C_{in} \times H \times W}$, Filters $F \in \mathbb{R}^{C_{out} \times C_{in} \times K \times K}$
**Output**: $Y \in \mathbb{R}^{C_{out} \times H \times W}$
**Initialize:** Hash function $h : \mathbb{R}^{(K+2)^2} \to \{0, 1, \ldots, 2^L - 1\}$

 1: **for** every patch $p$ **do**
 2:     HashCodes = [ ]                                   ▷ Create empty list for hash codes
 3:     **for** $i = 1, 2, \ldots, C_{in}$ **do**
 4:         $\boldsymbol{x}_i^{(p)} = $ Center(Flatten($X_i^{(p)}$))     ▷ Generate centered and flattened representation
 5:         HashCodes.Append($h(\boldsymbol{x}_i^{(p)})$)        ▷ Hash input representation and append code
 6:     **end for**
 7:     $\tilde{X}^{(p)} = $ MergeInput($X^{(p)}$, HashCodes)     ▷ Compute mean of channels in same bucket
 8:     $\tilde{F} = $ MergeFilters($F$, HashCodes)         ▷ Sum corresponding filter channels
 9:     $Y^{(p)} = \tilde{X}^{(p)} * \tilde{F}$
10: **end for**
11: **return** $Y$

---

Our HASTE module features two hyperparameters: the number of hyperplanes $L$ in the LSH scheme and the degree of sparsity $s$ in their normal vectors. Adjusting $L$ gives us a tractable trade-off between the compression ratio and approximation quality to the original convolution in the form of retained accuracy. This allows us to generate multiple model variants from one underlying base model, either focusing on low FLOPs or high accuracy. The normal vector sparsity $s$ does not require direct tuning and can easily be fixed across a dataset. Achlioptas (2003) and Li et al. (2006) provide initial values with theoretical guarantees. Our hyperparameter choices are discussed in Section 4.1.

## 4 EXPERIMENTS

In this section, we present results of our plug-and-play approach on standard CNN architectures in terms of FLOPs reduction as well as retained accuracy and give an overview of related methods. Firstly, we describe the setup of our experiments in detail. Then, we evaluate our proposed HASTE module on the CIFAR-10 (Krizhevsky, 2009) dataset for image classification and discuss the influence of the hyperparameter $L$. Lastly, we present results on the ImageNet ILSVRC 2012 (Russakovsky et al., 2015) benchmark dataset and discuss the scaling behavior of our method.

### 4.1 EXPERIMENT SETTINGS

For the experiments on CIFAR-10, we used pre-trained models provided by Phan (2021). On ImageNet, we use the trained models provided by PyTorch 2.0.0 (Paszke et al., 2019). Given a baseline model, we replace the regular non-strided convolutions with our HASTE module. For ResNet models (He et al., 2016), we do not include downsampling layers in our pruning scheme.

Depending on the dataset, we vary the degree of sparsity $s$ in the hyperplanes as well as at which layer we start pruning. As the CIFAR-10 dataset is less complex and features smaller latent spatial dimensions, we can increase the sparsity and prune earlier compared to models trained on ImageNet. For this reason, we set $s = 2/3$ on CIFAR-10 experiments as suggested by Achlioptas (2003), and start pruning VGG models (Simonyan & Zisserman, 2015) from the first convolution module and ResNet models from the first block after the max pooling operation. For experiments on ImageNet, we choose $s = 1/2$ to create random hyperplanes with less non-zero entries, leading to a more accurate hashing scheme. We prune VGG models starting from the third convolution module and ResNet / WideResNet models starting from the second layer. These settings compensate the lower degree of redundancy in latent feature maps of ImageNet models, especially in the early layers.

| Method | Dynamic | Restrictive Requirements | | |
| --- | --- | --- | --- | --- |
| | | Training | Fine-Tuning | Data Availability |
| SSL (Wen et al., 2016) | ✗ | ✗ | ✓ | ✓ |
| PFEC (Li et al., 2017) | ✗ | ✗ | ✓ | ✓ |
| LCCN (Dong et al., 2017) | ✓ | ✓ | ✗ | ✓ |
| FBS (Gao et al., 2019) | ✓ | ✓ | ✗ | ✓ |
| FPGM (He et al., 2019) | ✗ | ✓ | ✗ | ✓ |
| DynConv (Verelst & Tuytelaars, 2020) | ✓ | ✓ | ✗ | ✓ |
| DMCP (Xu et al., 2021) | ✓ | ✓ | ✗ | ✓ |
| DGNet (Li et al., 2021) | ✓ | ✓ | ✗ | ✓ |
| FTWT (Elkerdawy et al., 2022) | ✓ | ✓ | ✗ | ✓ |
| **HASTE (ours)** | ✓ | ✗ | ✗ | ✗ |

Table 1: Overview of related pruning approaches. While other methods require either fine-tuning or a specialized training procedure to achieve notable FLOPs reduction, our method is completely training-free and data-free.

After plugging in our HASTE modules, we directly evaluate the models on the corresponding test set using one NVIDIA Tesla T4 GPU, as no further fine-tuning or retraining is required. We follow common practice and report results on the validation set of the ILSVRC 2012 for models trained on ImageNet. Each experiment is repeated for three different random seeds to evaluate the effect of random hyperplane initialization. We report the mean top-1 accuracy after pruning and the mean FLOPs reduction compared to the baseline model as well as the standard deviation for both values.

Since, to the best of our knowledge, HASTE is the only approach that offers entirely data-free and dynamic model compression, we cannot give a direct comparison to similar work. For this reason, we resort to showing results of related channel pruning and dynamic gating approaches that feature specialized training or tuning routines. An overview of these methods is given in Table 1.

## 4.2 RESULTS ON CIFAR-10

For the CIFAR-10 dataset, we evaluate our method on ResNet18 and ResNet34 architectures as well as VGG11-BN, VGG13-BN, VGG16-BN and VGG19-BN. Results are presented in Figure 3a, while the full table of experiments is found in Appendix A.1. We also provide visualizations of pruned feature maps in Appendix D. Overall, our HASTE method achieves substantial reductions in the FLOPs requirement of tested networks. In particular, it reduces the computational cost of a ResNet34 by 46.72% with $L = 14$, while only losing 1.25 percentage points accuracy. This is achieved entirely without training, making the model less computationally expensive in an instant.

The desired ratio of cost reduction to accuracy loss can be adjusted on the fly by changing the hyperparameter $L$ across all HASTE modules simultaneously. Figure 3b shows how the relationship of targeted cost reduction and retained accuracy is influenced by the choice of $L$. Increased accuracy on the test set, achieved by increasing $L$, is directly related to less FLOPs reduction. For instance, we can vary the accuracy loss on ResNet34 between 2.89 ($L = 12$) and 0.38 ($L = 20$) percentage points to achieve 51.09% and 39.07% reduction in FLOPs, respectively.

We also give an overview of results from related approaches in Table 2. Although our method is not trained or fine-tuned on the dataset, it achieves comparable results to existing approaches which tailored their pruning scheme to the given data. Specifically, we outperform all other methods on VGG19-BN with 38.83% FLOPs reduction while retaining 92.32% accuracy, whereas the best trained approach (DMCP) achieves 34.14% cost reduction at 91.94% accuracy.

## 4.3 RESULTS ON IMAGENET

On the ImageNet benchmark dataset, we evaluate all available ResNet architectures including WideResNets as well as all VGG-BN models. Results are presented in Figure 4. A tabular overview of all experiments is given in Appendix A.2. In particular, we observe a positive scaling behavior of our method in Figure 4a, achieving up to 31.54% FLOPs reduction for a WideResNet101. When observing models of similar architecture, the potential FLOPs reduction grows with the number of parameters in a given model. We relate this to the fact that larger models typically exhibit more redundancies, which are then compressed by our module.

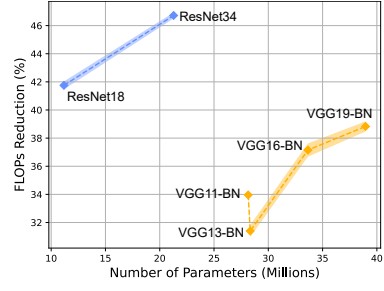

(a) Overview of CIFAR-10 results.

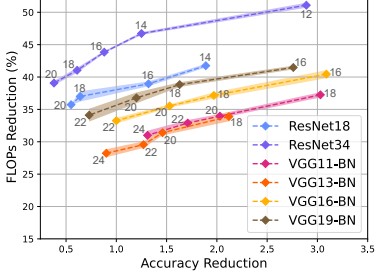

(b) Influence of hyperparameter $L$.

Figure 3: Results of our method on the CIFAR-10 dataset. (a) shows the achieved FLOPs reduction for all tested models, using $L = 14$ for ResNets and $L = 20$ for VGG-BN models. (b) depicts the influence of the chosen number of hyperplanes $L$ for LSH on compression rates and accuracy.

| Model | Method | Top-1 Accuracy (%) | | | FLOPs Reduction (%) | | |
|---|---|---|---|---|---|---|---|
| | | Baseline | Pruned | $\Delta$ | Tuning-Based | Training-Based | Data-Free |
| ResNet18 | PFEC* | 91.38 | 89.63 | 1.75 | 11.71 | - | - |
| | SSL* | 92.79 | 92.45 | 0.34 | 14.69 | - | - |
| | DMCP | 92.87 | 92.61 | 0.26 | - | 35.27 | - |
| | **Ours** ($L = 14$) | 93.07 | 91.18 ($\pm$0.38) | 1.89 | - | - | 41.75 ($\pm$0.28) |
| | **Ours** ($L = 20$) | 93.07 | 92.52 ($\pm$0.10) | 0.55 | - | - | 35.73 ($\pm$0.09) |
| VGG16-BN | PFEC* | 91.85 | 91.29 | 0.56 | 13.89 | - | - |
| | SSL* | 92.09 | 91.80 | 0.29 | 17.76 | - | - |
| | DMCP | 92.21 | 92.04 | 0.17 | - | 25.05 | - |
| | FTWT | 93.82 | 93.73 | 0.09 | - | 44.00 | - |
| | **Ours** ($L = 18$) | 94.00 | 92.03 ($\pm$0.21) | 1.97 | - | - | 37.15 ($\pm$0.47) |
| | **Ours** ($L = 22$) | 94.00 | 93.00 ($\pm$0.12) | 1.00 | - | - | 33.25 ($\pm$0.44) |
| VGG19-BN | PFEC* | 92.11 | 91.78 | 0.33 | 16.55 | - | - |
| | SSL* | 92.02 | 91.60 | 0.42 | 30.68 | - | - |
| | DMCP | 92.19 | 91.94 | 0.25 | - | 34.14 | - |
| | **Ours** ($L = 18$) | 93.95 | 92.32 ($\pm$0.35) | 1.63 | - | - | 38.83 ($\pm$0.36) |
| | **Ours** ($L = 22$) | 93.95 | 93.22 ($\pm$0.14) | 0.73 | - | - | 34.11 ($\pm$0.99) |

* Results taken from DMCP (Xu et al., 2021).

Table 2: Selected results on CIFAR-10. FLOPs Reduction denotes the percentage decrease of FLOPs after pruning compared to the base model.

Similar to He et al. (2018), we observe that models including pointwise convolutions are harder to prune than their counterparts which rely solely on larger filter kernels. This is particularly apparent in the drop in FLOPs reduction from ResNet34 to ResNet50. While the larger ResNet and WideResNet models with bottleneck blocks continue the scaling pattern, the introduction of pointwise convolutions momentarily dampens the computational cost reduction. Increasing the width of each convolutional layer benefits pruning performance, as is apparent with the results of WideResNet50 with twice the number of channels per layer as in ResNet50. While pointwise convolutions can achieve similar or even better compression ratios compared to $3 \times 3$ convolutions (see Figure 4b), the cost overhead of the hashing and merging steps is higher relative to the baseline.

When comparing the results to those seen on CIFAR-10, we note that our HASTE module achieves less compression on ImageNet classifiers. We directly relate this to the higher degree of complexity in the data. With a 100-fold increase in number of classes and roughly 26 times more training images than on CIFAR-10, the tested models store more information in latent feature maps, rendering them less redundant and therefore harder to compress. Methods that exploit the training data for extensively tuning their pruning scheme naturally achieve higher degrees of FLOPs reduction, as shown in Table 3. However, this is only possible when access to the data is granted. In contrast, our method offers significant reductions of computational cost in CNNs while being data-free, even scaling with larger model architectures.

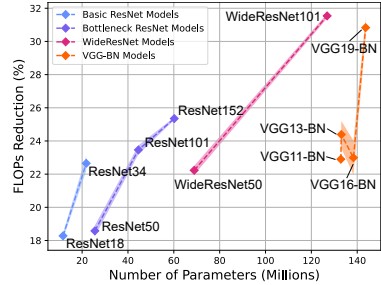

(a) Overview of ImageNet results.

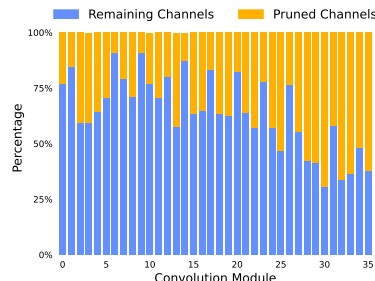

(b) Distribution of pruned channels in ResNet50.

Figure 4: Visualization of results on the ImageNet dataset. (a) depicts the relation of FLOPs reduction to number of parameters for all tested model architectures. Results are shown with $L = 16$ for basic ResNet models and $L = 28$ for bottleneck ResNets, $L = 32$ for WideResNets, and $L = 20$ for VGG-BN models. (b) shows the achieved compression rate per convolution module in a ResNet50, starting from the second bottleneck layer.

| Model | Method | Top-1 Accuracy (%) | | | FLOPs Reduction (%) | | |
|---|---|---|---|---|---|---|---|
| | | Baseline | Pruned | $\Delta$ | Tuning-Based | Training-Based | Data-Free |
| ResNet18 | LCCN | 69.98 | 66.33 | 3.65 | - | 34.60 | - |
| | DynConv* | 69.76 | 66.97 | 2.79 | - | 41.50 | - |
| | FPGM | 70.28 | 68.34 | 1.94 | 41.80 | - | - |
| | FPGM | 70.28 | 67.78 | 2.50 | - | 41.80 | - |
| | FBS | 70.71 | 68.17 | 2.54 | - | 49.49 | - |
| | FTWT | 69.76 | 67.49 | 2.27 | - | 51.56 | - |
| | **Ours** ($L = 16$) | 69.76 | 66.97 ($\pm 0.21$) | 2.79 | - | - | 18.28 ($\pm 0.19$) |
| | **Ours** ($L = 20$) | 69.76 | 68.64 ($\pm 0.56$) | 1.12 | - | - | 15.10 ($\pm 0.18$) |
| ResNet34 | PFEC | 73.23 | 72.09 | 1.14 | 24.20 | - | - |
| | LCCN | 73.42 | 72.99 | 0.43 | - | 24.80 | - |
| | FPGM | 73.92 | 72.54 | 1.38 | 41.10 | - | - |
| | FPGM | 73.92 | 71.79 | 2.13 | - | 41.10 | - |
| | FTWT | 73.30 | 72.17 | 1.13 | - | 47.42 | - |
| | DGNet | 73.31 | 71.95 | 1.36 | - | 67.20 | - |
| | **Ours** ($L = 16$) | 73.31 | 70.31 ($\pm 0.07$) | 3.00 | - | - | 22.65 ($\pm 0.45$) |
| | **Ours** ($L = 20$) | 73.31 | 72.06 ($\pm 0.05$) | 1.25 | - | - | 18.69 ($\pm 0.30$) |
| ResNet50 | FPGM | 76.15 | 74.83 | 1.32 | 53.50 | - | - |
| | FPGM | 76.15 | 74.13 | 2.02 | - | 53.50 | - |
| | DGNet | 76.13 | 75.12 | 1.01 | - | 67.90 | - |
| | **Ours** ($L = 28$) | 76.13 | 73.04 ($\pm 0.07$) | 3.09 | - | - | 18.58 ($\pm 0.33$) |
| | **Ours** ($L = 36$) | 76.13 | 74.77 ($\pm 0.10$) | 1.36 | - | - | 15.68 ($\pm 0.16$) |

* Results taken from DGNet (Li et al., 2021).

Table 3: Selected results on ImageNet. FLOPs Reduction denotes the percentage decrease of FLOPs after pruning compared to the base model.

## 5 CONCLUSION

While existing channel pruning approaches rely on training data to achieve notable reductions in computational cost, our proposed HASTE module removes restrictive requirements on data availability and compresses CNNs without any training steps. By employing a locality-sensitive hashing scheme for redundancy detection, we are able to drastically reduce the depth of latent feature maps and corresponding convolutional filters to significantly decrease the model's total FLOPs requirement. We empirically validate our claim through a series of experiments with a variety of CNN models and achieve compelling results on the CIFAR-10 and ImageNet benchmark datasets. We hope that our method acts as a first step in the direction of entirely data-free and training-free methods for the compression of convolutional architectures.

## ETHICS STATEMENT

Our work aims to enable compression of existing convolutional neural networks without having to spend additional computational resources on fine-tuning or retraining. Furthermore, our proposed HASTE module allows for pruning of pre-trained models without access to the training data. This has the potential to increase accessibility and usability of existing models by compressing them for usage on less powerful hardware, even when the training dataset is not publicly available. Furthermore, our method facilitates the employment of computationally cheaper models, reducing energy and carbon footprint induced by the network's inference.

However, advancements in the field of efficiency and accessibility of deep learning models, as well as AI in general, need to be considered under the aspect of their dual-use nature. While our work aims to enable broader access to large convolutional vision models, the same methodology could be applied in undesired scenarios. Specifically, the compression of vision models offers potential for misuse in military applications or mass surveillance, which raises ethical concerns regarding security and privacy. As authors, we distance ourselves from any application that may result in harm or negative societal impact.

## REPRODUCIBILITY STATEMENT

We aim to release our code upon publication. All pre-trained models, datasets and frameworks that were used in our experiments are publicly available. We give a detailed description of our experimental setup in Section 4.1. To increase reproducibility of our results, we performed every trial with three random seeds and averaged the results. Additionally, we provide a detailed overview of the FLOPs requirement of our method in Appendix B.1. We also discuss model design choices in Appendix C.

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

# A  FULL RESULTS

## A.1  CIFAR-10

Table 4 shows all results of our method on the CIFAR-10 dataset. All experiments were conducted as described in Section 4.1. We set $s = 2/3$ for all models. The last table column denotes the compression ratio $r$ achieved by our HASTE module, averaged over all patches, images and modules. We report the mean and standard deviation for every result, averaged over three random seeds.

| Model | $L$ | Top-1 Accuracy (%) | | | FLOPs | Average Compr. |
| | | Baseline | Pruned | $\Delta$ | Reduction (%) | Ratio $r$ (%) |
|---|---|---|---|---|---|---|
| ResNet18 | 14 | 93.07 | 91.18 ($\pm$0.38) | 1.89 | 41.75 ($\pm$0.28) | 57.07 ($\pm$0.26) |
| | 16 | 93.07 | 91.75 ($\pm$0.24) | 1.32 | 38.95 ($\pm$0.37) | 53.70 ($\pm$0.50) |
| | 18 | 93.07 | 92.43 ($\pm$0.16) | 0.64 | 37.00 ($\pm$0.79) | 51.38 ($\pm$0.94) |
| | 20 | 93.07 | 92.52 ($\pm$0.10) | 0.55 | 35.73 ($\pm$0.09) | 49.99 ($\pm$0.10) |
| ResNet34 | 12 | 93.34 | 90.45 ($\pm$0.47) | 2.89 | 51.09 ($\pm$0.38) | 62.31 ($\pm$0.43) |
| | 14 | 93.34 | 92.09 ($\pm$0.15) | 1.25 | 46.72 ($\pm$0.21) | 57.86 ($\pm$0.41) |
| | 16 | 93.34 | 92.46 ($\pm$0.15) | 0.88 | 43.84 ($\pm$0.16) | 54.75 ($\pm$0.24) |
| | 18 | 93.34 | 92.73 ($\pm$0.16) | 0.61 | 41.04 ($\pm$0.39) | 52.03 ($\pm$0.50) |
| | 20 | 93.34 | 92.96 ($\pm$0.06) | 0.38 | 39.07 ($\pm$0.32) | 50.09 ($\pm$0.36) |
| VGG11-BN | 18 | 92.39 | 89.36 ($\pm$0.01) | 3.03 | 37.25 ($\pm$0.40) | 48.47 ($\pm$0.86) |
| | 20 | 92.39 | 90.36 ($\pm$0.17) | 2.03 | 33.96 ($\pm$0.33) | 44.54 ($\pm$0.61) |
| | 22 | 92.39 | 90.68 ($\pm$0.16) | 1.71 | 32.89 ($\pm$0.47) | 43.81 ($\pm$0.39) |
| | 24 | 92.39 | 91.08 ($\pm$0.09) | 1.31 | 31.00 ($\pm$0.70) | 40.61 ($\pm$1.44) |
| VGG13-BN | 18 | 94.22 | 92.10 ($\pm$0.23) | 2.12 | 33.89 ($\pm$0.73) | 44.08 ($\pm$1.03) |
| | 20 | 94.22 | 92.76 ($\pm$0.21) | 1.46 | 31.40 ($\pm$0.42) | 41.47 ($\pm$0.61) |
| | 22 | 94.22 | 92.95 ($\pm$0.23) | 1.27 | 29.55 ($\pm$0.51) | 39.28 ($\pm$0.85) |
| | 24 | 94.22 | 93.32 ($\pm$0.10) | 0.90 | 28.23 ($\pm$0.62) | 37.26 ($\pm$0.40) |
| VGG16-BN | 16 | 94.00 | 90.91 ($\pm$0.49) | 3.09 | 40.44 ($\pm$0.62) | 51.22 ($\pm$0.78) |
| | 18 | 94.00 | 92.03 ($\pm$0.21) | 1.97 | 37.15 ($\pm$0.47) | 47.93 ($\pm$1.04) |
| | 20 | 94.00 | 92.47 ($\pm$0.42) | 1.53 | 35.54 ($\pm$0.25) | 45.91 ($\pm$0.63) |
| | 22 | 94.00 | 93.00 ($\pm$0.12) | 1.00 | 33.25 ($\pm$0.44) | 43.40 ($\pm$1.19) |
| VGG19-BN | 16 | 93.95 | 91.19 ($\pm$0.42) | 2.76 | 41.47 ($\pm$0.29) | 52.55 ($\pm$0.71) |
| | 18 | 93.95 | 92.32 ($\pm$0.35) | 1.63 | 38.83 ($\pm$0.36) | 49.70 ($\pm$0.32) |
| | 20 | 93.95 | 92.75 ($\pm$0.33) | 1.20 | 36.78 ($\pm$0.83) | 47.66 ($\pm$0.72) |
| | 22 | 93.95 | 93.22 ($\pm$0.14) | 0.73 | 34.11 ($\pm$0.99) | 45.23 ($\pm$1.19) |

Table 4: Full results of our method on the CIFAR-10 dataset.

## A.2  IMAGENET

Table 5 shows all results of our method on the ImageNet dataset. All experiments were conducted as described in Section 4.1. We set $s = 1/2$ for all models. The last table column denotes the compression ratio $r$ achieved by our HASTE module, averaged over all patches, images and modules. We report the mean and standard deviation for every result, averaged over three random seeds.

# B  FLOPS SAVINGS AND MEMORY BENCHMARKS

## B.1  DERIVING THEORETICAL FLOPS SAVINGS

In this section, we provide an exact computation of the FLOPs saving which are achieved by our HASTE module. To accurately determine the improvement, we first need to count the FLOPs of

| Model | $L$ | Top-1 Accuracy (%) | | | FLOPs Reduction (%) | Average Compr. Ratio $r$ (%) |
|---|---|---|---|---|---|---|
| | | Baseline | Pruned | $\Delta$ | | |
| ResNet18 | 16 | 69.76 | 66.97 ($\pm$0.21) | 2.79 | 18.28 ($\pm$0.19) | 35.70 ($\pm$0.37) |
| | 18 | 69.76 | 67.87 ($\pm$0.09) | 1.89 | 16.53 ($\pm$0.47) | 32.49 ($\pm$0.88) |
| | 20 | 69.76 | 68.63 ($\pm$0.06) | 1.12 | 15.10 ($\pm$0.18) | 29.88 ($\pm$0.34) |
| | 22 | 69.76 | 68.87 ($\pm$0.06) | 0.88 | 14.41 ($\pm$0.15) | 28.66 ($\pm$0.28) |
| | 24 | 69.76 | 69.00 ($\pm$0.13) | 0.76 | 13.78 ($\pm$0.17) | 27.56 ($\pm$0.31) |
| | 26 | 69.76 | 69.20 ($\pm$0.02) | 0.56 | 13.15 ($\pm$0.31) | 26.46 ($\pm$0.57) |
| ResNet34 | 16 | 73.31 | 70.31 ($\pm$0.07) | 3.00 | 22.65 ($\pm$0.45) | 36.93 ($\pm$0.58) |
| | 18 | 73.31 | 71.57 ($\pm$0.04) | 1.74 | 20.08 ($\pm$0.32) | 33.20 ($\pm$0.39) |
| | 20 | 73.31 | 72.07 ($\pm$0.05) | 1.25 | 18.69 ($\pm$0.30) | 31.16 ($\pm$0.42) |
| ResNet50 | 28 | 76.13 | 73.04 ($\pm$0.07) | 3.09 | 18.58 ($\pm$0.33) | 39.25 ($\pm$0.54) |
| | 36 | 76.13 | 74.77 ($\pm$0.10) | 1.36 | 15.68 ($\pm$0.16) | 34.59 ($\pm$0.15) |
| ResNet101 | 28 | 77.37 | 74.90 ($\pm$0.23) | 2.47 | 23.46 ($\pm$0.27) | 40.60 ($\pm$0.40) |
| | 36 | 77.37 | 76.17 ($\pm$0.15) | 1.20 | 20.16 ($\pm$0.25) | 36.15 ($\pm$0.56) |
| ResNet152 | 28 | 78.31 | 76.07 ($\pm$0.20) | 2.24 | 25.35 ($\pm$0.08) | 42.06 ($\pm$0.12) |
| | 36 | 78.31 | 77.39 ($\pm$0.07) | 0.92 | 21.62 ($\pm$0.04) | 36.88 ($\pm$0.12) |
| WideResNet50 | 32 | 78.47 | 76.01 ($\pm$0.13) | 2.46 | 22.23 ($\pm$0.27) | 46.37 ($\pm$0.19) |
| WideResNet101 | 32 | 78.85 | 75.95 ($\pm$0.29) | 2.90 | 31.54 ($\pm$0.14) | 51.07 ($\pm$0.27) |
| VGG11-BN | 20 | 70.37 | 69.64 ($\pm$0.07) | 0.73 | 22.90 ($\pm$0.56) | 29.85 ($\pm$0.37) |
| | 28 | 70.37 | 70.08 ($\pm$0.07) | 0.29 | 18.97 ($\pm$0.70) | 25.13 ($\pm$0.81) |
| VGG13-BN | 20 | 71.59 | 70.25 ($\pm$0.38) | 1.34 | 24.39 ($\pm$0.82) | 32.09 ($\pm$1.04) |
| | 28 | 71.59 | 71.04 ($\pm$0.15) | 0.55 | 21.04 ($\pm$1.07) | 27.65 ($\pm$1.38) |
| VGG16-BN | 20 | 73.36 | 72.19 ($\pm$0.19) | 1.17 | 22.99 ($\pm$1.03) | 29.92 ($\pm$0.79) |
| | 28 | 73.36 | 72.87 ($\pm$0.06) | 0.49 | 18.74 ($\pm$0.52) | 24.87 ($\pm$0.59) |
| VGG19-BN | 20 | 74.22 | 71.50 ($\pm$0.81) | 2.72 | 30.83 ($\pm$0.63) | 36.97 ($\pm$0.54) |
| | 28 | 74.22 | 73.25 ($\pm$0.19) | 0.97 | 27.51 ($\pm$0.16) | 32.81 ($\pm$0.13) |

Table 5: Full results of our method on the ImageNet dataset.

the underlying baseline convolution module. Since a convolution operation requires an almost equal amount of additions and multiplications, we can derive its FLOPs count from the number of performed multiply-accumulate operations, or MACs, by multiplying with a factor of two. Given a module with $C_{in}$ input and $C_{out}$ output channels, kernel size $K$ and input spatial size $H \times W$, the total amount of FLOPs for a regular convolution operation amounts to

$$\text{FLOPs}_{\text{Regular}} = 2 \cdot H \cdot W \cdot K^2 \cdot C_{in} \cdot C_{out}. \tag{5}$$

We are assuming that the input is padded such that the spatial resolution remains the same after the convolution. Otherwise, we would simply subtract the overhang of the convolutional filter on all sides from the spatial resolution. Similarly, it is assumed that the stride is set to one. In any other case, we can determine the new output sizes $H', W'$ when using stride $> 2$ and use them to replace $H, W$ in Equation 5. Furthermore, we assume the use of non-dilated kernels.

For our HASTE module, the overall cost is determined by the sum of FLOPs for the individual components: centering, hashing, feature map merging, filter merging and reduced convolution. In the LSH step, each individual channel at patch position $p$ is flattened into a $(K + 2)^2$-dimensional vector. The resulting $C_{in}$ many vectors are then centered by subtracting their mean across the channel dimension. This ensures that the LSH hyperplanes, which are also centered at the coordinate origin, are able to partition the channel vectors in a meaningful way. Due to the overlap of two pixels at the border between neighboring patches, we get $(H + P_H)/K \cdot (W + P_W)/K$ patches in total, where $P_H$ and $P_W$ denote the respective padding of input height and width. Therefore, the centering

step introduces a computational overhead of

$$\text{FLOPs}_{\text{Centering}} = 2 \cdot \frac{H + P_H}{K} \cdot \frac{W + P_W}{K} \cdot C_{in} \cdot (K + 2)^2 \,, \tag{6}$$

When using random projections, the hashing process itself consists of computing $C_{in} \cdot L$ dot products between the flattened channel vectors and each hyperplane. As the hashing step is performed per patch $p$, we have a total of $(H + P_H)/K \cdot (W + P_W)/K \cdot C_{in} \cdot L$ dot products. Each dot product uses $2 \cdot K^2$ FLOPs when choosing the hyperplane normal vectors $\mathbf{v}_l$ with entries sampled from $\mathcal{N}(0, 1)$. However, when using sparse hyperplanes as discussed in Section 3.1, this cost reduces in two ways. Firstly, the number of operations performed drops by the factor $s \in (0, 1)$, the degree of sparsity specifying the expected ratio of non-zero entries to zero entries in each hyperplane normal vector. This is equivalent to multiplying the total cost with factor $1 - s$. Secondly, as the sparse hyperplane dot product only requires additions and no multiplication, the amount of FLOPs is halved. Overall, this results in a total cost of

$$\text{FLOPs}_{\text{Hashing}} = \frac{H + P_H}{K} \cdot \frac{W + P_W}{K} \cdot C_{in} \cdot L \cdot (K + 2)^2 \cdot (1 - s). \tag{7}$$

As for the merging of input feature map channels, computing the mean of all redundant channels accounts for $(K + 2)^2 \cdot (C_{in} \cdot r + m)$ FLOPs, where $m$ denotes the average number of buckets where channels are merged and $r$ denotes the average compression ratio over every patch position $p$ across all input feature maps. As this step is repeated for each filter position, we get

$$\text{FLOPs}_{\text{MergeFMs}} = \frac{H + P_H}{K} \cdot \frac{W + P_W}{K} \cdot (K + 2)^2 \cdot (C_{in} \cdot r + m). \tag{8}$$

The merging step for the convolutional filters requires $C_{in} \cdot r \cdot K^2$ FLOPs per filter, as only additions are performed. This results in an overall cost of

$$\text{FLOPs}_{\text{MergeFilters}} = \frac{H + P_H}{K} \cdot \frac{W + P_W}{K} \cdot C_{out} \cdot C_{in} \cdot r \cdot K^2. \tag{9}$$

When computing the convolution on the merged input features with aggregated filters, we can perform $K^2$ reduced convolutions for every filter with kernel size $K$ inside the patches of size $(K + 2) \times (K + 2)$. Skipping convolutions where the center of the kernel lies outside the original input's spatial size, we get a total of

$$\text{FLOPs}_{\text{ReducedConv}} = 2 \cdot H \cdot W \cdot K^2 \cdot C_{in} \cdot (1 - r) \cdot C_{out}. \tag{10}$$

floating-point operations for computing the reduced convolution. One can easily see now that the deciding factor for computational cost savings is the reduction of input channel size, resulting in a reduction of FLOPs by the factor $(1 - r)$ when comparing only the convolution operations.

Overall, we can state the cost of the HASTE module by summing the cost of its individual parts:

$$\begin{aligned} \text{FLOPs}_{\text{HASTE}} = {} & \text{FLOPs}_{\text{Centering}} + \text{FLOPs}_{\text{Hashing}} + \text{FLOPs}_{\text{MergeFMs}} \\ & + \text{FLOPs}_{\text{MergeFilters}} + \text{FLOPs}_{\text{ReducedConv}} \,. \end{aligned} \tag{11}$$

## B.2 PRUNING POINTWISE CONVOLUTIONS

A special case of the convolution operation appears when $K = 1$. These $1 \times 1$ convolutions are commonly used for downsampling or upsampling of the channel dimension before and after parameter-heavy convolutions with larger kernel sizes, or after a depth-wise convolutional layer. However, as the kernel resolution changes to a single pixel, each input pixel generates exactly one output pixel in the spatial domain. As there is no reduction in spatial resolution when performing $1 \times 1$ convolutions, we do not require the $3 \times 3$ patches that rasterize the input to be overlapping. Hence, we pad the input in such a way that each side is divisible by 3 and use non-overlapping patches.

## B.3 MEMORY BENCHMARKS

In this section, we provide memory benchmarks for our proposed HASTE module. As our method offers a dynamic, input-dependent compression of latent feature maps and the corresponding filters, we have to store the underlying base model's weights throughout inference. For this reason, we

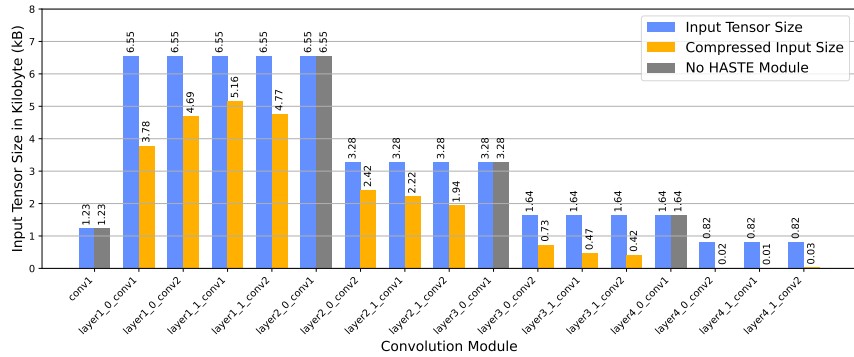

Figure 5: Memory requirements for input tensors in ResNet18 ($L = 14$) on CIFAR-10.

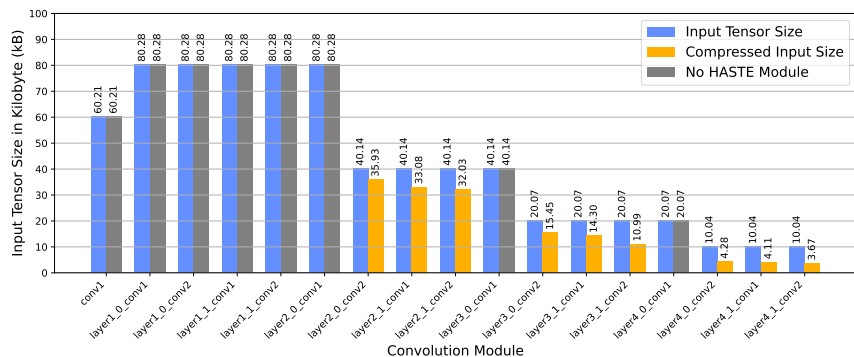

Figure 6: Memory requirements for input tensors in ResNet18 ($L = 16$) on ImageNet.

do not reduce the model's overall size as is the case for static pruning methods. However, the compression of latent feature maps is essential to avoid high memory bus utilization, which is critical for employment on embedded hardware (Vogel et al., 2019). The HASTE module compresses the channel dimension of an input tensor before the convolution operation, allowing for efficient storage of the reduced input.

Let $r = 1 - (\tilde{C}_{in}/C_{in}) \in (0, 1)$ be the compression ratio of an input tensor's size as defined in Section 3.3. As the memory requirement of a a tensor is the product of the number of elements it contains and the memory size of each such element, a compression of the tensor's size is equivalent to the compression of its memory requirement by the same amount. Therefore, the average compression ratio $r$ presented in Tables 4 and 5 can directly be understood as the average compression rate of input tensor memory requirements.

Figures 5 and 6 visualize the relative compression of latent input sizes per convolutional module in a ResNet18 architecture, evaluated on both the CIFAR-10 test set and ImageNet validation set. We observe that, especially in later layers, the input tensor size can be reduced drastically, reducing memory requirement by more than half. It is worth noting that Figure 4b from Section 4.3 displays similar information, as the compression ratio can directly be translated to memory savings.

## C   ABLATION STUDIES

### C.1   IMPORTANCE OF LOCALITY-SENSITIVE HASHING

As a simple check to determine whether our LSH scheme finds meaningful similarities in the channel dimension of latent feature maps, we test a naive random grouping of channels. For this experiment, we use a ResNet18 on the CIFAR-10 dataset, following the same experiment setup as described in Section 4.1. To ensure a fair comparison between randomized grouping and LSH, we extract the number of channels to be merged per hash bucket from the hashing scheme. We utilize this to

aggregate the exact same amount of randomly selected channels, such that the compression ratio $r$ remains similar across both approaches. Results are shown in Table 6.

| Model | Channel Selection | $L$ | Top-1 Accuracy (%) | | | FLOPs Reduction (%) | Average Compr. Ratio $r$ (%) |
|---|---|---|---|---|---|---|---|
| | | | Baseline | Pruned | $\Delta$ | | |
| ResNet18 | LSH | 14 | 93.07 | 91.18 ($\pm$0.38) | 1.89 | 41.75 ($\pm$0.28) | 57.07 ($\pm$0.26) |
| | | 16 | 93.07 | 91.75 ($\pm$0.24) | 1.32 | 38.95 ($\pm$0.37) | 53.70 ($\pm$0.50) |
| | | 18 | 93.07 | 92.43 ($\pm$0.16) | 0.64 | 37.00 ($\pm$0.79) | 51.38 ($\pm$0.94) |
| | | 20 | 93.07 | 92.52 ($\pm$0.10) | 0.55 | 35.73 ($\pm$0.09) | 49.99 ($\pm$0.10) |
| | Random | 14 | 93.07 | 11.12 ($\pm$0.34) | 81.95 | 44.30 ($\pm$0.49) | 59.11 ($\pm$0.55) |
| | | 16 | 93.07 | 11.03 ($\pm$0.36) | 82.04 | 40.91 ($\pm$0.68) | 54.83 ($\pm$0.93) |
| | | 18 | 93.07 | 11.17 ($\pm$0.25) | 81.90 | 38.72 ($\pm$0.94) | 52.03 ($\pm$1.12) |
| | | 20 | 93.07 | 12.37 ($\pm$0.40) | 80.70 | 37.10 ($\pm$0.20) | 50.02 ($\pm$0.40) |

Table 6: Comparison of LSH to random channel grouping on CIFAR-10.

It is obvious that a random grouping of channels does not maintain the model's accuracy when compressing the channel dimension of latent features and filters. In fact, when randomly selecting channels to compress, the accuracy is only slightly better than random guessing on CIFAR-10. Note that for the random approach, the value of $L$ only affects the compression ratio $r$, as we simulate the same number of channels to be merged as with the LSH scheme.

With decreasing compression ratio, the random grouping starts to get slightly more accurate as less latent information is arbitrarily aggregated. Also, the random approach achieves a marginally higher reduction in FLOPs for similar compression ratios, as the overhead for hashing and the centering of channel vectors is omitted. However, the random grouping is not able to retain usable accuracy on the test set. This validates that our proposed LSH scheme does find meaningful clusters of channels in latent feature maps.

## C.2 Influence of Hyperplane Sparsity

In this section, we provide an ablation for the hyperparameter $s$, which determines the degree of sparsity in the hyperplane's normal vectors used in our LSH scheme. For this reason, we evaluate our proposed HASTE module on the CIFAR-10 and ImageNet datasets as described in Section 4.1. For both datasets, we fix the model choice to a ResNet18 architecture and try various values for $s$ over a fixed amount of hyperplanes $L$. We denote the usage of dense Gaussian hyperplanes with normal vector entries sampled from $\mathcal{N}(0, 1)$ by "None". The special case of $s = 0$ occurs when using densely filled hyperplane normal vectors, but restricting their entries to being chosen from the set $\{-1, 1\}$ with equal probability. For a more detailed introduction to the sparsity parameter $s$, see Section 3.1.

The results of our experiments for different choices of $s$ are presented in Tables 7 and 8. We also visualize the results in Figures 7 and 8, where we plot the reduction in accuracy ($\Delta$) per percentage point of FLOPs reduction. This ratio helps us to compare the results across different sparsity settings. A lower ratio is better, as it implies less accuracy is lost for constant FLOPs reduction, or more FLOPs are saved for constant accuracy reduction. It determines the amount of percentage points in accuracy lost for every 1 percentage point of FLOPs saved.

**CIFAR-10.** By observing the mean accuracy reduction per FLOPs reduction ratio in Figure 7, we conclude that the performance of our proposed HASTE module is not strongly dependent on the choice of $s$. For the ResNet18 model architecture, dense Gaussian projections perform as well as sparse hyperplanes with $s \in \{0.2, 0.\overline{3}, 0.5, 0.\overline{6}\}$. However, note that Gaussian projections require multiplications to be performed at runtime to compute the hash codes. This is not the case for sparse projections, where a few hardware-friendly additions suffice. As the ratio displayed in Figure 7 only considers overall FLOPs savings and not solely FLOPs used in the hashing step, it is still beneficial to choose sparse hyperplanes for an efficient implementation. When increasing the sparsity ratio further to $s = 0.8$, we start to loose more accuracy per FLOPs saved. For this reason, we chose the highest sparsity setting $s = 0.\overline{6}$ before the performance starts to degrade for our experiments

| Model | $L$ | $s$ | Top-1 Accuracy (%) | | | FLOPs Reduction (%) |
|---|---|---|---|---|---|---|
| | | | Baseline | Pruned | $\Delta$ | |
| ResNet18 | 14 | None | 93.07 | 91.58 ($\pm$0.54) | 1.49 | 35.28 ($\pm$0.22) |
| | | 0 | 93.07 | 90.36 ($\pm$0.49) | 2.71 | 39.87 ($\pm$0.56) |
| | | 0.2 | 93.07 | 91.35 ($\pm$0.15) | 1.72 | 39.39 ($\pm$0.17) |
| | | 0.$\overline{3}$ | 93.07 | 91.64 ($\pm$0.20) | 1.43 | 38.84 ($\pm$0.43) |
| | | 0.5 | 93.07 | 91.30 ($\pm$0.26) | 1.77 | 39.64 ($\pm$0.29) |
| | | 0.$\overline{6}$ | 93.07 | 91.18 ($\pm$0.38) | 1.89 | 41.75 ($\pm$0.28) |
| | | 0.8 | 93.07 | 88.53 ($\pm$0.11) | 4.54 | 46.19 ($\pm$0.39) |
| | 16 | None | 93.07 | 92.31 ($\pm$0.06) | 0.76 | 32.43 ($\pm$0.59) |
| | | 0 | 93.07 | 90.66 ($\pm$0.81) | 2.41 | 37.82 ($\pm$0.52) |
| | | 0.2 | 93.07 | 92.25 ($\pm$0.14) | 0.82 | 35.89 ($\pm$0.25) |
| | | 0.$\overline{3}$ | 93.07 | 91.99 ($\pm$0.34) | 1.08 | 36.35 ($\pm$0.28) |
| | | 0.5 | 93.07 | 92.19 ($\pm$0.09) | 0.88 | 37.05 ($\pm$0.23) |
| | | 0.$\overline{6}$ | 93.07 | 91.75 ($\pm$0.24) | 1.32 | 38.95 ($\pm$0.37) |
| | | 0.8 | 93.07 | 91.19 ($\pm$0.06) | 1.88 | 42.52 ($\pm$0.26) |
| | 18 | None | 93.07 | 92.56 ($\pm$0.09) | 0.51 | 30.49 ($\pm$0.19) |
| | | 0 | 93.07 | 91.73 ($\pm$0.20) | 1.34 | 35.52 ($\pm$0.33) |
| | | 0.2 | 93.07 | 92.23 ($\pm$0.17) | 0.84 | 34.33 ($\pm$0.36) |
| | | 0.$\overline{3}$ | 93.07 | 92.31 ($\pm$0.12) | 0.76 | 34.51 ($\pm$0.37) |
| | | 0.5 | 93.07 | 92.47 ($\pm$0.36) | 0.60 | 35.19 ($\pm$0.44) |
| | | 0.$\overline{6}$ | 93.07 | 92.43 ($\pm$0.16) | 0.64 | 37.00 ($\pm$0.79) |
| | | 0.8 | 93.07 | 91.59 ($\pm$0.21) | 1.48 | 40.66 ($\pm$0.49) |
| | 20 | None | 93.07 | 92.74 ($\pm$0.14) | 0.33 | 28.21 ($\pm$0.29) |
| | | 0 | 93.07 | 92.00 ($\pm$0.08) | 1.07 | 34.04 ($\pm$0.04) |
| | | 0.2 | 93.07 | 92.55 ($\pm$0.03) | 0.52 | 32.82 ($\pm$0.21) |
| | | 0.$\overline{3}$ | 93.07 | 92.31 ($\pm$0.12) | 0.76 | 34.51 ($\pm$0.37) |
| | | 0.5 | 93.07 | 92.58 ($\pm$0.19) | 0.49 | 32.49 ($\pm$0.58) |
| | | 0.$\overline{6}$ | 93.07 | 92.52 ($\pm$0.10) | 0.55 | 35.73 ($\pm$0.09) |
| | | 0.8 | 93.07 | 92.04 ($\pm$0.16) | 1.03 | 39.02 ($\pm$0.24) |

Table 7: Comparison of different hyperparameter choices for $s$ on CIFAR-10. The "None" setting for $s$ denotes the use of dense Gaussian projections, where the hyperplane normal vector's entries are sampled from $\mathcal{N}(0,1)$. Setting $s = 0$ implies normal vectors which are densely filled with entries from $\{-1, 1\}$.

on CIFAR-10. Nonetheless, the usage of dense projections might be considered, as they offer a direct link between collision probability and vector cosine similarity (Li et al., 2006). Furthermore, we note that for $s = 0$, the performance is notably worse than for its neighboring hyperparameter settings. We relate this to the fact that hyperplane normal vectors, which are densely filled with entries from $\{-1, 1\}$, offer less variety than their sparsely filled counterparts. Instead of $3^d$ possible $d-$dimensional random vectors, the dense case only allows for $2^d$ distinct possibilities. This increases the chance of hash collisions for elements that further apart and thus not necessarily redundant.

**ImageNet.** On the ImageNet benchmark, we observe a similar pattern as on the CIFAR-10 dataset in Figure 8. The ResNet18 architecture seems to be relatively robust to the choice of hyperparameter $s$, offering comparable performance across a range of sparsity settings and also using dense Gaussian projections. However, as above, we favor the sparse hyperplanes to avoid multiplications when computing the hash codes of input channels. Empirically, we found that the performance of larger model architectures, particularly of those that utilize pointwise convolutions, degrades more rapidly on ImageNet when using a high degree of sparsity such as $s = 2/3$. For this reason, we chose to evaluate all architectures on ImageNet with $s = 0.5$. However, higher hyperplane sparsity settings can be valid for specific model architectures.

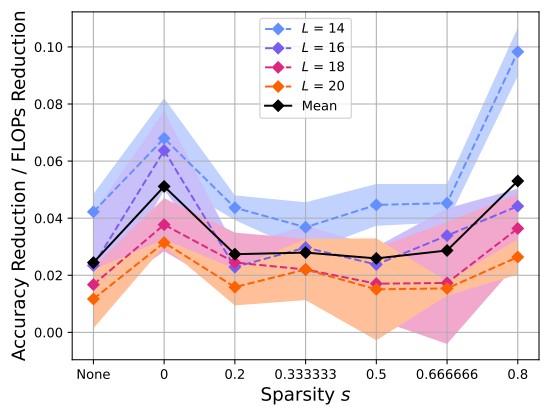

Figure 7: Influence of sparsity $s$ on the CIFAR-10 dataset. We plot the ratio of accuracy reduction ($\Delta$) per percentage point of FLOPs reduction for each hyperparameter choice.

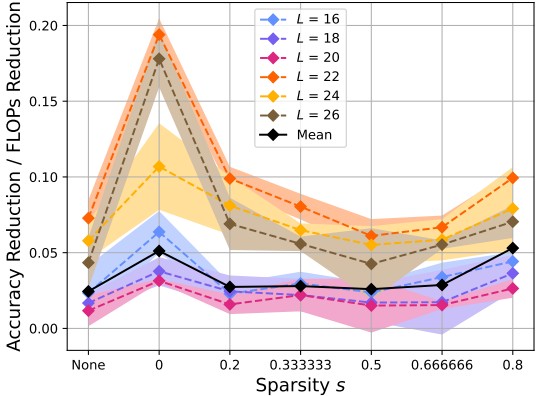

Figure 8: Influence of sparsity $s$ on the ImageNet dataset. We plot the ratio of accuracy reduction ($\Delta$) per percentage point of FLOPs reduction for each hyperparameter choice.

| Model | $L$ | $s$ | Top-1 Accuracy (%) | | | FLOPs Reduction (%) |
|---|---|---|---|---|---|---|
| | | | Baseline | Pruned | $\Delta$ | |
| ResNet18 | 16 | None | 69.76 | 67.28 ($\pm$0.02) | 2.47 | 16.71 ($\pm$0.31) |
| | | 0 | 69.76 | 64.99 ($\pm$0.37) | 4.77 | 18.55 ($\pm$0.20) |
| | | 0.2 | 69.76 | 66.40 ($\pm$0.08) | 3.36 | 17.91 ($\pm$0.15) |
| | | $0.\overline{3}$ | 69.76 | 66.59 ($\pm$0.18) | 3.17 | 18.23 ($\pm$0.04) |
| | | 0.5 | 69.76 | 66.97 ($\pm$0.21) | 2.79 | 18.28 ($\pm$0.19) |
| | | $0.\overline{6}$ | 69.76 | 66.91 ($\pm$0.16) | 2.84 | 19.48 ($\pm$0.37) |
| | | 0.8 | 69.76 | 65.63 ($\pm$0.26) | 4.13 | 21.92 ($\pm$0.21) |
| | 18 | None | 69.76 | 67.99 ($\pm$0.02) | 1.77 | 14.71 ($\pm$0.24) |
| | | 0 | 69.76 | 66.01 ($\pm$0.20) | 3.75 | 17.01 ($\pm$0.29) |
| | | 0.2 | 69.76 | 67.48 ($\pm$0.20) | 2.28 | 16.13 ($\pm$0.17) |
| | | $0.\overline{3}$ | 69.76 | 67.74 ($\pm$0.16) | 2.01 | 16.35 ($\pm$0.37) |
| | | 0.5 | 69.76 | 67.87 ($\pm$0.09) | 1.89 | 16.53 ($\pm$0.47) |
| | | $0.\overline{6}$ | 69.76 | 67.87 ($\pm$0.12) | 1.89 | 17.51 ($\pm$0.21) |
| | | 0.8 | 69.76 | 66.61 ($\pm$0.07) | 3.15 | 20.05 ($\pm$0.15) |
| | 20 | None | 69.76 | 68.63 ($\pm$0.03) | 1.12 | 13.51 ($\pm$0.30) |
| | | 0 | 69.76 | 67.64 ($\pm$0.89) | 2.11 | 15.40 ($\pm$0.49) |
| | | 0.2 | 69.76 | 67.97 ($\pm$0.02) | 1.79 | 15.08 ($\pm$0.20) |
| | | $0.\overline{3}$ | 69.76 | 68.46 ($\pm$0.07) | 1.30 | 15.09 ($\pm$0.15) |
| | | 0.5 | 69.76 | 68.63 ($\pm$0.06) | 1.12 | 15.10 ($\pm$0.18) |
| | | $0.\overline{6}$ | 69.76 | 68.33 ($\pm$0.05) | 1.43 | 16.63 ($\pm$0.38) |
| | | 0.8 | 69.76 | 67.36 ($\pm$0.11) | 2.39 | 18.89 ($\pm$0.14) |
| | 22 | None | 69.76 | 68.85 ($\pm$0.04) | 0.91 | 12.50 ($\pm$0.15) |
| | | 0 | 69.76 | 66.88 ($\pm$0.18) | 2.88 | 14.85 ($\pm$0.16) |
| | | 0.2 | 69.76 | 68.36 ($\pm$0.14) | 1.40 | 14.15 ($\pm$0.11) |
| | | $0.\overline{3}$ | 69.76 | 68.61 ($\pm$0.10) | 1.14 | 14.18 ($\pm$0.12) |
| | | 0.5 | 69.76 | 68.87 ($\pm$0.06) | 0.88 | 14.41 ($\pm$0.15) |
| | | $0.\overline{6}$ | 69.76 | 68.72 ($\pm$0.08) | 1.04 | 15.57 ($\pm$0.12) |
| | | 0.8 | 69.76 | 68.01 ($\pm$0.11) | 1.75 | 17.61 ($\pm$0.08) |
| | 24 | None | 69.76 | 69.10 ($\pm$0.06) | 0.66 | 11.40 ($\pm$0.14) |
| | | 0 | 69.76 | 68.30 ($\pm$0.73) | 1.46 | 13.66 ($\pm$0.39) |
| | | 0.2 | 69.76 | 68.66 ($\pm$0.12) | 1.09 | 13.44 ($\pm$0.26) |
| | | $0.\overline{3}$ | 69.76 | 68.89 ($\pm$0.06) | 0.87 | 13.41 ($\pm$0.06) |
| | | 0.5 | 69.76 | 69.00 ($\pm$0.13) | 0.76 | 13.78 ($\pm$0.17) |
| | | $0.\overline{6}$ | 69.76 | 68.90 ($\pm$0.07) | 0.86 | 14.69 ($\pm$0.20) |
| | | 0.8 | 69.76 | 68.42 ($\pm$0.16) | 1.33 | 16.81 ($\pm$0.46) |
| | 26 | None | 69.76 | 69.29 ($\pm$0.02) | 0.47 | 10.83 ($\pm$0.18) |
| | | 0 | 69.76 | 67.30 ($\pm$0.17) | 2.46 | 13.82 ($\pm$0.26) |
| | | 0.2 | 69.76 | 68.88 ($\pm$0.04) | 0.88 | 12.75 ($\pm$0.22) |
| | | $0.\overline{3}$ | 69.76 | 69.06 ($\pm$0.04) | 0.70 | 12.54 ($\pm$0.06) |
| | | 0.5 | 69.76 | 69.20 ($\pm$0.02) | 0.56 | 13.15 ($\pm$0.31) |
| | | $0.\overline{6}$ | 69.76 | 68.97 ($\pm$0.21) | 0.79 | 14.28 ($\pm$0.04) |
| | | 0.8 | 69.76 | 68.64 ($\pm$0.03) | 1.12 | 15.90 ($\pm$0.17) |

Table 8: Comparison of different hyperparameter choices for $s$ on ImageNet. The "None" setting for $s$ denotes the use of dense Gaussian projections, where the hyperplane normal vector's entries are sampled from $\mathcal{N}(0, 1)$. Setting $s = 0$ implies normal vectors which are densely filled with entries from $\{-1, 1\}$.

| Model | Patch Size | Top-1 Accuracy (%) | | | FLOPs Reduction (%) | Average Compr. Ratio $r$ (%) |
|---|---|---|---|---|---|---|
| | | Baseline | Pruned | $\Delta$ | | |
| ResNet18 | $5 \times 5$ | 69.76 | 68.63 ($\pm$0.06) | 1.12 | 15.10 ($\pm$0.18) | 30.05 ($\pm$0.24) |
| | $7 \times 7$ | 69.76 | 68.87 ($\pm$0.02) | 0.89 | 9.69 ($\pm$0.18) | 18.37 ($\pm$0.32) |
| | $9 \times 9$ | 69.76 | 69.43 ($\pm$0.03) | 0.33 | 3.72 ($\pm$0.15) | 7.08 ($\pm$0.26) |

Table 9: Comparison of different patch sizes on ImageNet.

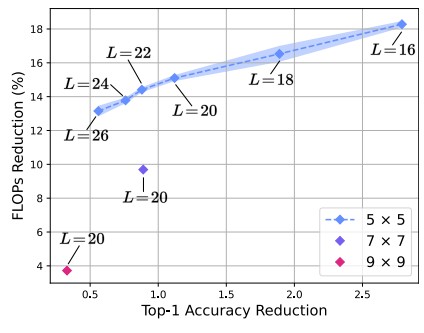

(a) Influence of patch size on FLOPs.

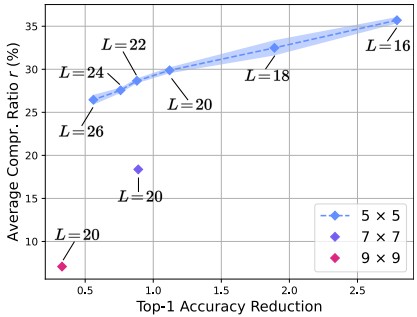

(b) Influence of patch size on compression.

Figure 9: Evaluation of trade-offs between compression and accuracy for different patch sizes.

## C.3 INFLUENCE OF PATCH SIZE

Our HASTE module requires the input patch to be larger than the filter kernel in the spatial dimension to benefit from repeated convolutions with shallower inputs and filters. Given a filter kernel size $K$, a natural choice are patches of size $(K + 2) \times (K + 2)$. This allows us to perform nine reduced convolutions per patch, which drastically reduces the overhead of filter and feature map merging. Furthermore, smaller patch sizes naturally lead to more redundancies in the channel dimension due to having less context per patch compared to larger sizes.

To evaluate different patch sizes and their effectiveness, we tested a ResNet18 on ImageNet equipped with our HASTE modules in the same setting as described in Section 4.1. As all convolution modules in ResNet18 use a kernel size $K = 3$, we used patch sizes $5 \times 5$, $7 \times 7$ and $9 \times 9$ for this experiment. Furthermore, we set $L = 20$ for all trials. Results are presented in Table 9. Additionally, we visualize the different trade-offs between achieved model compression and retained accuracy in Figure 9.

We observe that our chosen patch size $(K + 2) \times (K + 2)$ offers the best trade-off between FLOPs reduction and model accuracy. Larger patch sizes exhibit fewer redundancies in the channel dimension and therefore offer reduced compression ratios and corresponding FLOPs reduction. While this benefits the model's accuracy, it does so disproportionately to the reduction in FLOPs when compared to the baseline $5 \times 5$ patches. By varying the hyperparameter $L$, we can create various model variants using $5 \times 5$ patches that outperform larger patch sizes.

## D VISUALIZATIONS

### D.1 CHANNEL COMPRESSION FREQUENCY

In this section, we visualize the frequency of merge operations between input channels across the entire CIFAR-10 test dataset. Figures 10, 11, 12 and 13 show a heatmap of the relative frequency of merges for each pairing of input channels, generated from a ResNet18 architecture with $L = 14$. The axes label each input channel by its index. The maximum relative frequency of 1 would imply that a pair of channels was merged consistently for every patch and every single image in the test set. Note that the heatmaps are symmetric along their main diagonal, which is filled with zeroes, as a single channel is never merged with itself.

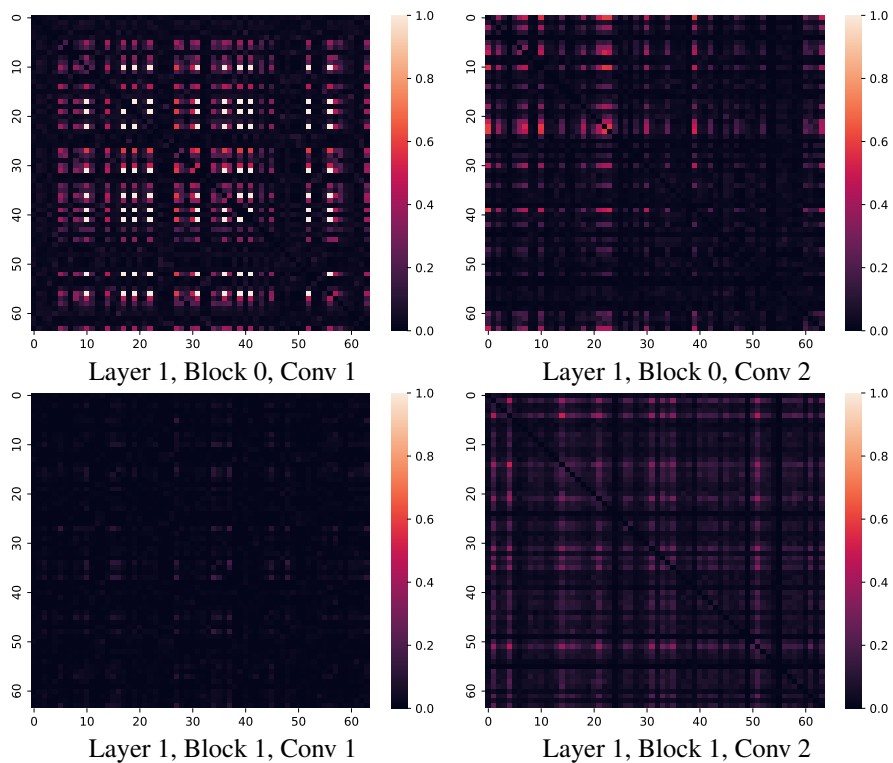

Figure 10: Compression Heatmap for Layer 1 of ResNet18.

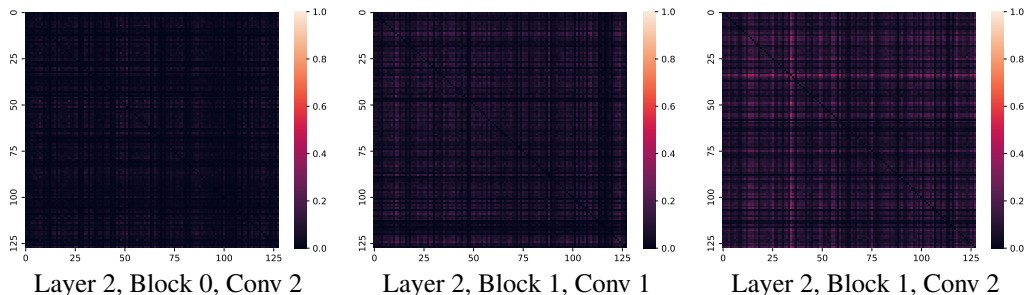

Figure 11: Compression Heatmap for Layer 2 of ResNet18.

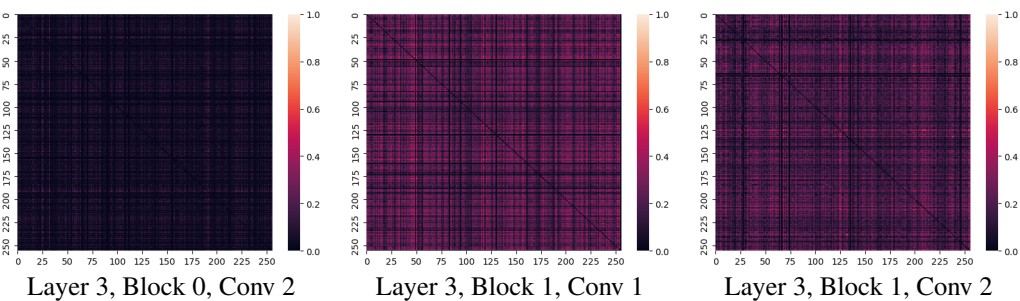

Figure 12: Compression Heatmap for Layer 3 of ResNet18.

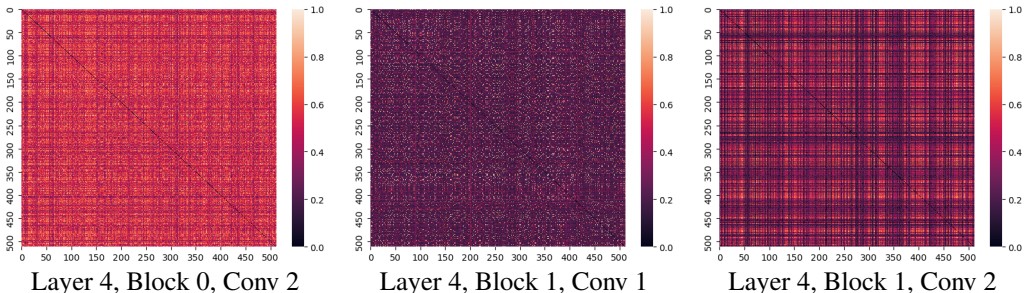

Figure 13: Compression Heatmap for Layer 4 of ResNet18.

The average intensity of each heatmap gives a rough estimate on the compression ratio $r$ that is achieved in every layer, which is particularly noticeable when comparing Figures 11, 12 and 13. The last layer of the ResNet model offers the most redundancies, leading to the highest compression ratios. This is consistent with our findings in Figure 5 and Figure 4b. In Figure 10. We also notice a prominent pattern for the first convolution in layer 1. The bright grid cells imply a structural redundancy in the first convolutional filter, as the corresponding channels were merged across data samples and varying patch positions.

## D.2 FEATURE MAP COMPRESSION

To get an intuitive understanding of the merge operation for redundant feature map channels as described in Section 3.3, we provide visualizations of the latent features before and after the merging step. For this purpose, we track the latent input feature maps and the resulting compression for two different images from the CIFAR-10 dataset, observed in a HASTE module in the first layer of a ResNet18 model using $L = 14$ and $s = 2/3$.

Figures 14, 15, 16 and 17 illustrate our proposed HASTE module. Firstly, the spatial dimension of the input feature map is split into patches. Then, for each patch (marked by a red border), we detect redundancies by hashing the flattened vector representation $x_i^{(p)}$ of each input channel. We visualize this by coloring the border of patches with identical hash codes with identical colors. Patches that do not have a colored outline did not match any other hash code, and are therefore left unchanged. All patches in the same hash bucket, represented by shared outline color, are then merged to a singular channel by computing their mean. Therefore, the overall input channel dimension is reduced to $\tilde{C}_{in} < C_{in}$. The same channels are then merged in the corresponding convolutional filters, allowing as to perform a convolution using fewer floating-point operations.

Note that the compression ratio $r = 1 - (\tilde{C}_{in}/C_{in}) \in (0, 1)$ changes not only depending on the input image, but on the amount of redundancies found in each individual patch. The comparison of Figure 14 and 15 reveal an interesting property of our proposed HASTE module: Patches that contain little class-specific information, such as the background, can be compressed to a much higher degree than patches that contain relevant information for the classification task. This is attributable to the learned filter kernels in pre-trained models, which activate more intensely and more diversely in regions with higher semantic information content. This leads to numerous near-zero channels being compressed into a single representation. Nonetheless, many redundancies are also found by observing similar activation patterns throughout different non-zero channels.

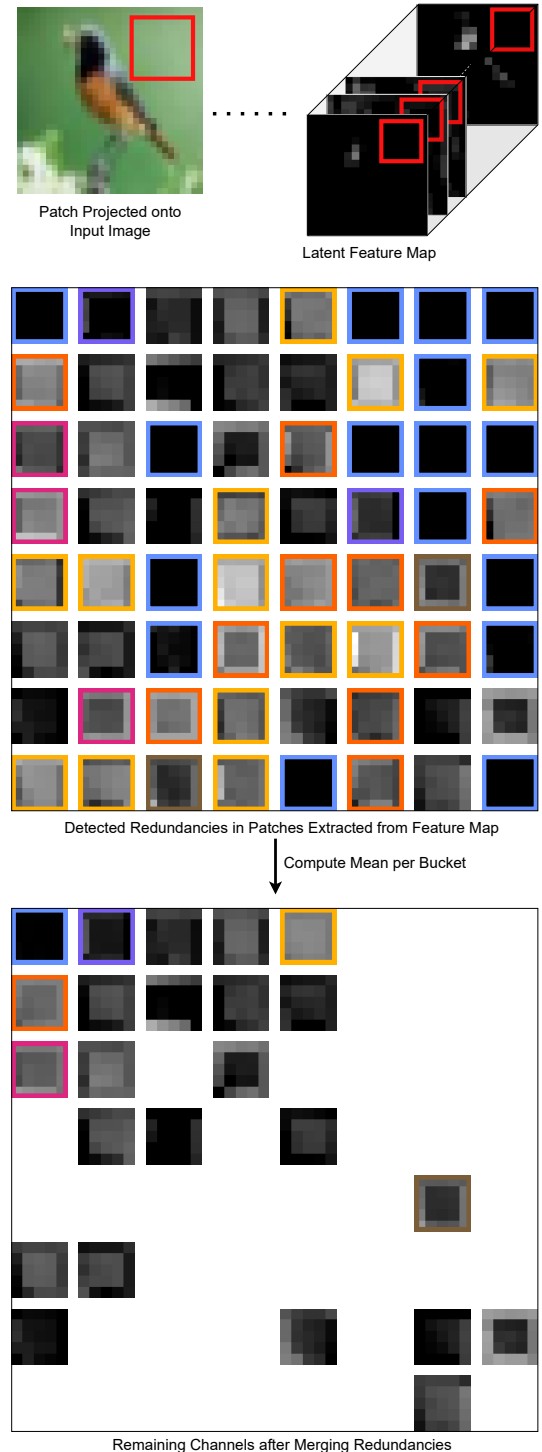

Figure 14: Visualization of the input channel compression performed by the HASTE module. The observed patch is marked as a red square on the input feature maps. All 64 channels of this patch are then plotted in an $8 \times 8$ grid. Patches with identical hash codes receive identical outline colors and are averaged by taking their mean. Patches with no matching hash code are left unchanged. Here, we reduce $C_{in} = 64$ to $\tilde{C}_{in} = 24$, which gives us a compression ratio of $r = 62.50\%$.

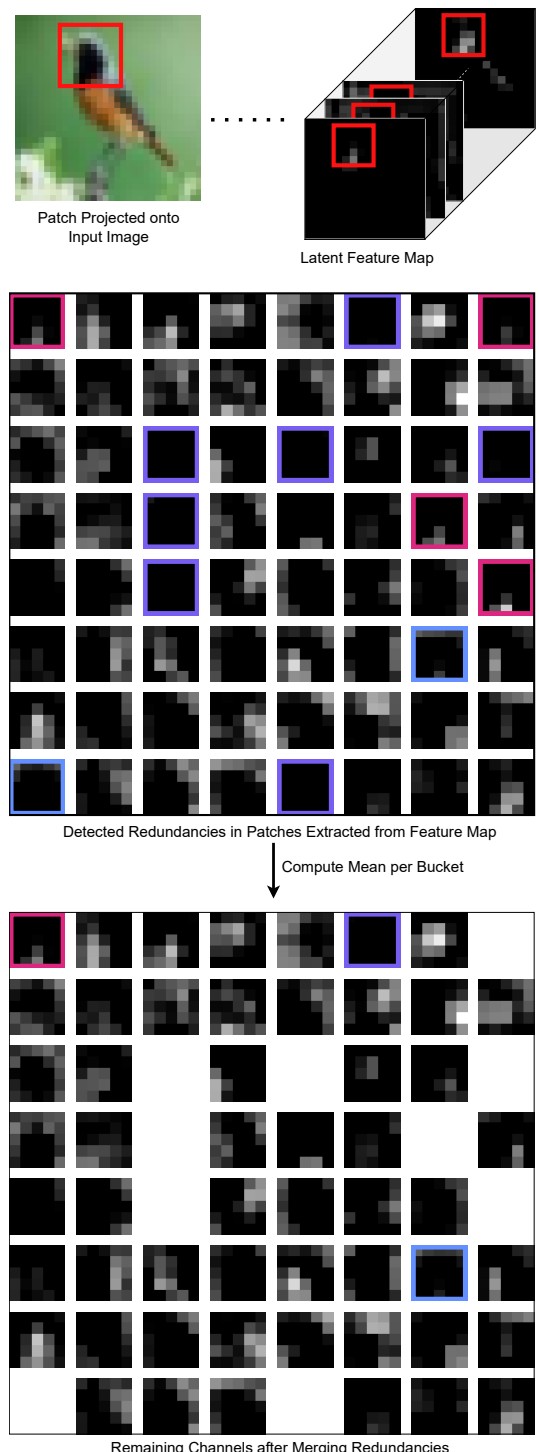

Figure 15: Visualization of the input channel compression performed by the HASTE module. The observed patch is marked as a red square on the input feature maps. All 64 channels of this patch are then plotted in an $8 \times 8$ grid. Patches with identical hash codes receive identical outline colors and are averaged by taking their mean. Patches with no matching hash code are left unchanged. Here, we reduce $C_{in} = 64$ to $\tilde{C}_{in} = 54$, which gives us a compression ratio of $r = 15.63\%$.

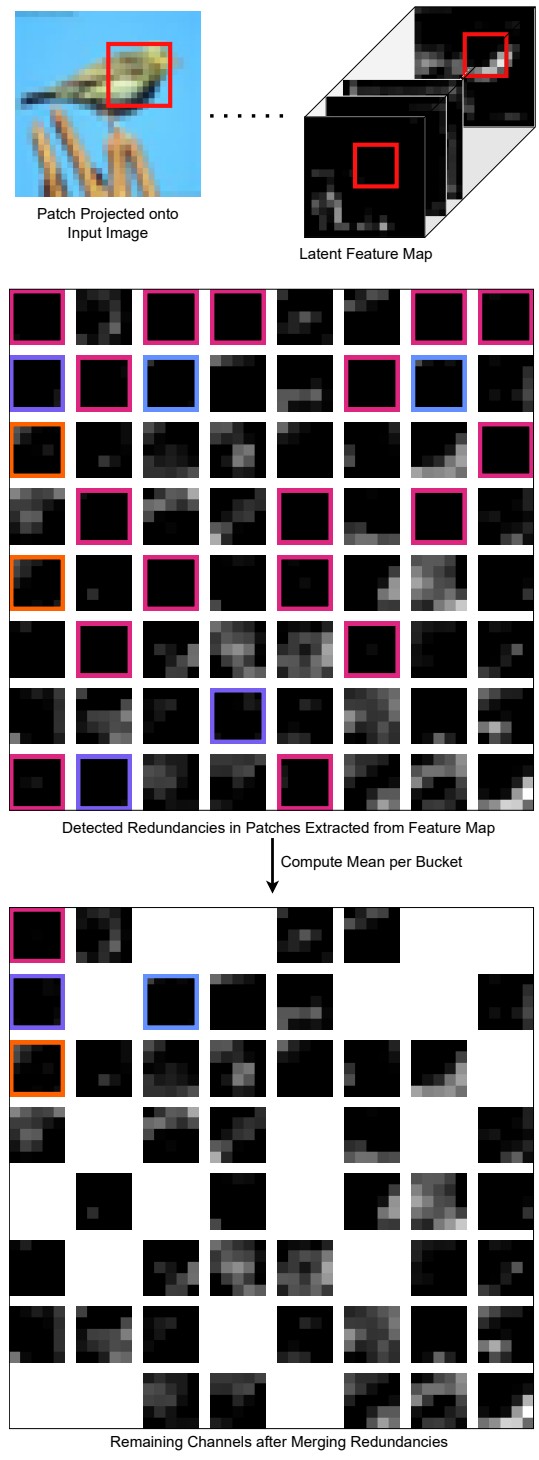

Figure 16: Visualization of the input channel compression performed by the HASTE module. The observed patch is marked as a red square on the input feature maps. All 64 channels of this patch are then plotted in an $8 \times 8$ grid. Patches with identical hash codes receive identical outline colors and are averaged by taking their mean. Patches with no matching hash code are left unchanged. Here, we reduce $C_{in} = 64$ to $\tilde{C}_{in} = 44$, which gives us a compression ratio of $r = 31.25\%$.

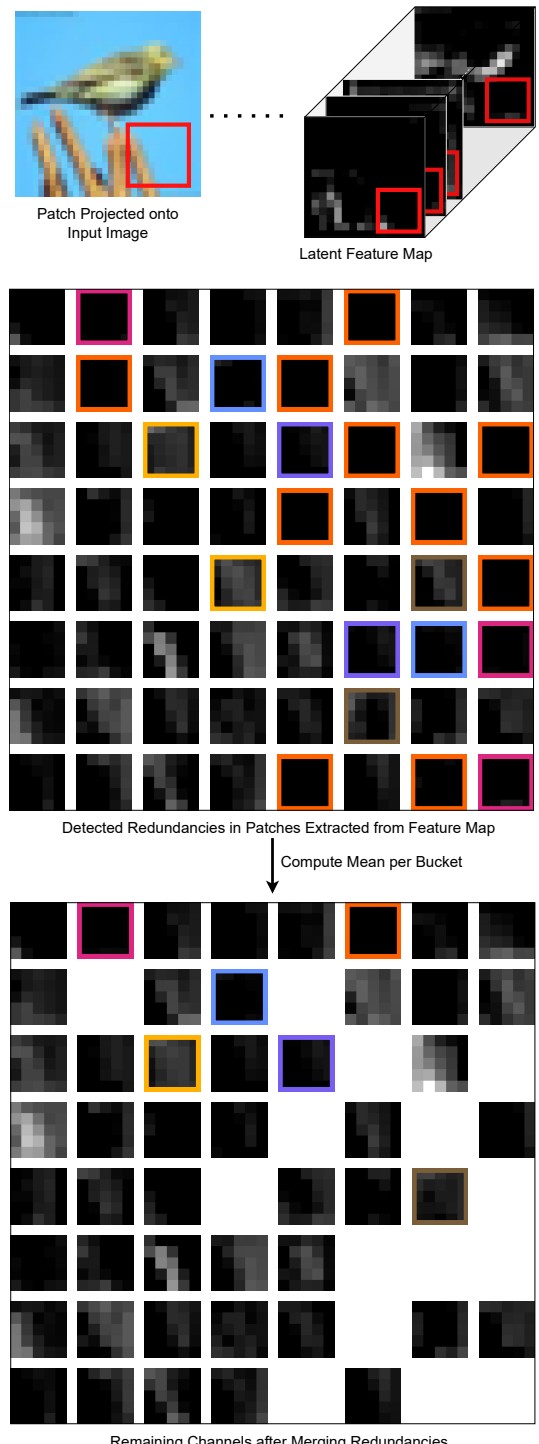

Figure 17: Visualization of the input channel compression performed by the HASTE module. The observed patch is marked as a red square on the input feature maps. All 64 channels of this patch are then plotted in an $8 \times 8$ grid. Patches with identical hash codes receive identical outline colors and are averaged by taking their mean. Patches with no matching hash code are left unchanged. Here, we reduce $C_{in} = 64$ to $\tilde{C}_{in} = 49$, which gives us a compression ratio of $r = 23.43\%$.

