# OpenReview forum: "Instant Complexity Reduction in CNNs using Locality-Sensitive Hashing"
_ICLR.cc/2024/Conference — Submitted to ICLR 2024_

### Official Review · Reviewer_AoyU · 2023-10-14

**Soundness:** 3 good
**Presentation:** 3 good
**Contribution:** 1 poor
**Rating:** 3
**Confidence:** 3

**Summary:**

The paper proposes removing redundant channels, by hashing. Critically, the method focuses on the post-training scenario and is data-free. The authors evaluate on ImageNet and CIFAR.

**Strengths:**

- The paper conveys the motivation, method, and experimental results decently well. The critical elements are present -- ImageNet results, some ablations, and a clear methods.

**Weaknesses:**

- The motivation is clearly written but there's a gap between that and the method. In the introduction, with the paragraph starting from "However, the application of existing work is restricted..." clearly motivates post-training methods that are data-free. That makes sense, and your method falls in that category. However, what's less clear is why hashing in particular improves over other methods in this category of data-free, post-training methods. Why is hashing the magic solution that defeats all other post-training methods?
- For example, another way to leverage the linearity of the convolution (per first few sentences in sec 3.3) is to simply use kmeans. What does hashing buy us that kmeans could not? In fact, maybe kmeans is a reasonable baseline to include?
- Another question that could be answered with ablations: Why square patches? Structured pruning support on hardware isn't usually in this patch form. A100s for example support 2:4 sparsity.

**Questions:**

- nit: Table 3's organization of the right 3 columns are a bit weird. Maybe have a column for FLOPS reduction and then separately have a "type" column (tuning, training, or data-free). Or, if you really want to use more space, have three columns with checkmarks. It's hard as-is to compare results/numbers. (Hrm, but this is already in Table 1. Why repeat this in Table 3?)

---

> ### Author Response · Authors · 2023-11-16
> **Response to concerns and questions**
>
> We thank the reviewer for their time to provide feedback and comments on our work. We are encouraged to find that the presentation of our work is well-received. We address the reviewer's concerns and answer open questions below.
> ## Concerns:
> **Why is hashing chosen over other methods for pruning in a data-free and post-training setting?** Our proposed approach utilizes locality-sensitive hashing (LSH) primarily due to its ability to run dynamically during model inference. This distinguishes it from existing approaches for data-free and post-training pruning, which are static and perform pre-processing on the trained model's weights to generate a fixed pruned model. To the best of our knowledge, this makes the HASTE module the first method that reduces model complexity, in particular, floating-point operations (FLOPs)
> - in a data-free manner
> - without requiring any pre-processing of the model's weights
> - dynamically (dependent on the module's input) during model inference.
>
> and as such, acts as a plug-and-play replacement for convolutional modules. To cluster redundant information in latent feature maps dynamically, we require an approximate nearest neighbor search that is fast to run online, has limited FLOPs overhead and is able to adapt to the current input feature map. LSH fulfills these strict requirements and has shown its potential in previous work, as stated in Section 2 of our paper.
>
> **Could $k$-means be an alternative to hashing?** Yes, it is possible to replace hashing by $k$-means clustering. However, in our scenario, it is not applicable. The reason for this is two-fold:
> - Firstly, with $k$-means, we would need to fix the number of clusters $k$, enforcing a predetermined compression ratio onto each latent input. This can introduce large errors when compressing non-redundant feature channels when $k$ was set too high. Similarly, we might miss potential for pruning when there are more than $k$ clusters in the features.
> - Secondly, the computational overhead of $k$-means is infeasible when considering dynamic pruning at runtime. Our proposed locality-sensitive hashing (LSH) approach has time complexity $O(L \cdot n \cdot d \cdot (1-s))$, where $L$ is the number of hyperplanes, $s\in(0,1)$ their degree of sparsity, $n$ is the number of vectors to be hashed and $d$ is their dimensionality. However, $k$-means clustering has time complexity $O(I \cdot n \cdot d \cdot k)$, where $k$ is the number of cluster centers and $I$ is the number of required iterations. We would require multiple iterations $I$ and set $k$ to the number of distinct channels we want to maintain after pruning. This quickly exceeds the cost of hashing (where $14 \leq L \leq 36$) and would require extensive hyperparameter testing to set custom values for $k$ at each layer. This is not required in our LSH approach, where layer-wise dynamic pruning ratios are possible due to variable numbers of hash collisions using the same fixed $L$ throughout all layers.
>
> **Why square patches? Does this interfere with hardware support for sparsity patterns?** The choice of square patches is motivated in a similar way to the choice of square filter kernels for performing the convolution operation. Using square patches allows us to easily rasterize the square input image while preserving redundancies present in local neighborhoods of pixels. As discussed in Appendix A.5.2, larger patches exhibit less redundancies and allow for less compression of the input feature map. We believe that choosing square patches is a natural way to leverage redundancies in the input without making further assumptions on the shape of these features, as would be the case for rectangular patches. Given prior knowledge on the shape of latent feature maps however, the use of other patch shapes could be beneficial.
> Regarding hardware support for sparsity patterns, we have not considered tailoring our approach to any specific hardware. While the A100 cards supports a 2:4 sparsity pattern, this works purely on weight-level. As our approach is structurally removing entire channels from the convolution and thus working on tensor-level, we believe that it would not benefit from this specific pruning pattern. While we aim to keep our approach broadly applicable, custom optimizations are an interesting possibility to explore in future work.
>
> ## Questions / Remarks:
> **Organization of Table 3 makes it difficult to compare results. Maybe change to one "Type" column or use checkmarks?** We found that the current table format offered the clearest distinction of the three compared pruning approaches. We deemed it necessary to clearly distinguish our results from related work at a glance. As our method is fundamentally different by offering dynamic pruning without training or fine-tuning, the obtained results are not directly comparable.  To further improve readability, we might explore other formats and update Table 3 in the revised version of the paper.

---

> ### Comment · Reviewer_AoyU · 2023-12-03
> **Thanks for the clarifications**
>
> Thanks to the authors for their clarifications; I have two concerns mainly, based on the clarifications:
>
> 1. No latency reduction: The critical issue is that dynamically-changing pruning patterns can't be (easily) leveraged by hardware for latency reductions -- even though the paper reports FLOP reductions, this is at best a poor proxy for latency and at worst a misleading statistic. (Non-contiguous filters in memory are enough to completely reverse latency benefits for pruning)
>
> 2. No storage reduction: Since weights aren't pruned in advance, there aren't savings on disk for model weights. In turn, there also aren't any memory savings for the parameters.
>
> So "dynamic pruning patterns" is not so much an overlooked idea as it is an infeasible idea. If the pruning patterns had some structure in the innermost loop (e.g., along rows, and not in spatial squares), it *might* be possible... but still unlikely. As a result, I maintain my current score, due to the impracticality of the method in netting us downstream benefits.
>
> (And you're certainly correct k-means couldn't be run at inference time; makes sense to dismiss that suggestion)

---

### Official Review · Reviewer_JSr2 · 2023-11-01

**Soundness:** 2 fair
**Presentation:** 3 good
**Contribution:** 1 poor
**Rating:** 3
**Confidence:** 4

**Summary:**

The authors propose to study input clustering for convolutional neural networks. For a given convolutional layer, the method proposes to dynamically cluster spatial features in order to reduce the number of computations performed at inference. The resulting method, dubbed HASTE is data-free and parameter-free. The authors evaluate the method on ConvNets trained for Cifar10 and ImageNet.

**Strengths:**

The method is fairly well introduced and provides good results on existing convolutional neural networks such as ResNet or VGG. The study of hashlists is often overlooked in the compression literature as compared to pruning and uniform quantization when it can, in fact, provide significant benefits.

In my opinion, the most considerable strength of this piece of research is the interest on intermediate features compression.

**Weaknesses:**

I have multiple concerns regarding this work. First, in terms of novelty, the idea of merging the inputs based on redundancies obtained by hashing and only computing relevant operations is not new [1]. In particular, the differences between HASTE and [1] are marginal, and the empirical evidence suggests that [1] outperforms HASTE.

Second, the authors claim that the proposed HASTE method is parameter-free (first bullet point in the introduction) but then proceed to define two hyperparameters : L and s. Furthermore, these hyperparameters are dataset specific, and we do not have an automatic way to find them.

Third, the benchmarks are lacking. While I understand that not all researchers have access to the same compute, this research is performed in a data-free context. This means that the authors should be able to benchmark on more recent and larger scale models. From my perspective, this is not sufficient for a venue such as ICLR.

Finally, the authors claim that the method induces a limited overhead. It seems to me that this overhead is only benchmarked in terms of FLOPs. I understand that FLOPs as a metric has benefits (simple to benchmark and is hardware-agnostic) but it should be made explicit that we are not considering latency nor memory footprint, which limits the interest and impact of the method. On most hardware, having multiple sequential operations at a low level added to a network will significantly slow it down. For instance, it is well known that BN layers have very little overhead in terms of FLOPs while removing them can divide by 2 the latency on CPUs and GPUs.

[1] Yvinec, Edouard, et al. "Red++: Data-free pruning of deep neural networks via input splitting and output merging." IEEE Transactions on Pattern Analysis and Machine Intelligence 45.3 (2022): 3664-3676.

**Questions:**

My questions are listed in the form of concerns above. In summary, I would appreciate it if the authors could: 1, detail the difference with [1], 2, remove the parameter-free claim 3, benchmark on an LLM or a diffusion model and 4 provide evidence or comments on the latency and memory footprint impact of the proposed method.

---

> ### Author Response · Authors · 2023-11-16
> **Response to concerns and questions**
>
> We thank the reviewer for the effort spent to provide feedback on our work. We are encouraged to find that our motivation to compress latent feature maps for efficient computations is well-received. We address the reviewer's concerns and answer open questions below.
> ## Concerns:
> **The method used by the authors is not new and was already used in [1], and the difference between both papers is marginal. Results suggest that [1] outperforms the proposed HASTE approach.** We disagree with this statement. While both papers share the aspect of using hashing to detect and merge redundant information in CNNs, a detailed analysis reveals fundamental differences.
> - Firstly, Red++ [1] offers static pruning, where the model is compressed once and fixed in a pruned state for inference. In contrast, our HASTE module offers dynamic pruning, which adapts the compression to the input tensor during inference and requires no pre-processing of the model. This allows us to change the model's total FLOPs requirement instantly at runtime. We can therefore react to varying availability of hardware resources. The pruning steps in Red++ "require a few minutes in the worst case scenario, never exceeding half an hour" [1, Appendix C], which is infeasible for online inference. When the hashed model weights are not stored throughout inference, an even more expensive re-hashing of the base model is necessary, which can take hours for large models trained on ImageNet [1, Appendix C]. In contrast, the HASTE module does not require adjustments when the underlying model's weights are changed, as the compression is performed online on latent feature maps. This allows for applications in a federated learning or online learning setting with continuous weight updates. We acknowledge the need to clarify this distinction in our paper and will add a corresponding segment in the revised version.
> - Secondly, the mechanisms used to compress information differ substantially in both works. In [1], the authors quantize ("hash") the individual weights of the baseline model to introduce redundancies in the weight matrices. We do not alter the model's weights and instead perform hashing using signed random projections, which map entire feature channels to a lower-dimensional space, where we find similarities between them. Red++ then structurally removes redundant components of the convolutional filters based on the quantized weights and generates a static pruned network. In comparison, we keep all filters and their weights throughout the network and compress them on-the-fly, based on redundancies in the current input. This allows us to maintain the model's expressivity. We achieve compression by directly comparing hash codes of input feature map channels and averaging their information content.
> - While the results presented in [1] are impressive, we argue that in the instant pruning setting considered in our paper does not allow for a direct comparison of both approaches. We are limited by the compute budget available at runtime. The authors of [1] are not subject to such limitations and make use of extensive pre-processing of the model's weights, which can take multiple hours and is computationally expensive. Nonetheless, we hypothesize that both approaches can be combined to improve the results. Unfortunately, the authors of [1] do not provide publicly available code.
>
> **The authors claim that their approach is parameter-free, but introduce two hyperparameters. Can this claim be removed?** This is correct, and we understand how this statement is misleading. To clarify this and avoid any confusion, we will remove this claim in the revised version of our paper.
>
> **The provided benchmarks are lacking.** We provide a wide range of results for different models and additionally show scaling behavior within different model architectures. Nonetheless, our future work includes extensive testing of our proposed method on a variety of architectures and tasks. We hope to include more experiments in the revised paper.
>
> **The proposed approach is only benchmarked with regard to floating-point operations. Can the authors comment on latency and memory requirements?** We are aware of this concern and aim to benchmark our model in terms of memory requirement and latency to provide a more complete analysis in the revised version. Figure 2 shows the potential for memory reduction.
>
> ## Questions:
> **Can the authors test their method on LLMs or a diffusion model?** We focus our research on convolutional architectures due to their wide-spread usage in embedded applications. The extension of our method to LLMs or diffusion models would require considerable effort, and is out of scope for this paper.
>
> [1] Yvinec, Edouard, et al. "Red++: Data-free pruning of deep neural networks via input splitting and output merging." IEEE Transactions on Pattern Analysis and Machine Intelligence 45.3 (2022): 3664-3676.

---

### Official Review · Reviewer_fMDs · 2023-11-02

**Soundness:** 3 good
**Presentation:** 3 good
**Contribution:** 3 good
**Rating:** 6
**Confidence:** 5

**Summary:**

The paper proposes HASTE for computation reduction at inference time for CNNs. The key idea is that the in a convolution layer, there might be similar channels forming 'clusters'. The paper uses locality sensitive hashing (LSH) to efficiently identify similar channels in a same LSH bucket, and compute the convolution only once for the averaged vector in each hash bucket. Experiments is conducted to show the trade-off between complexity and accuracy. HASTE is able to considerably reduce the FLOPs and maintain a reasonable model utility.

**Strengths:**

Applying classic ML techniques to deep learning is a promising direction and hashing is one good example. There have been many attempts in recent years trying to use hashing to accelerate deep learning training and inference, and the paper continues this line of work. I appreciate the simple and intuitive idea of HASTE.

The writing is clear in general, and the paper is easy to follow. The empirical results are convincing; many model architectures are tested.

**Weaknesses:**

1. The LSH used for the HASTE module is a variant of SimHash, but it lacks rigorous justification/theory. Different from SimHash using dense Gaussian projections, the paper used very sparse random projection (VSRP) to reduce the complexity. However, SimHash has a strict mapping from the collision probability to the cosine similarity, but signed VSRP does not (asymptotic analysis under some data assumptions might be doable, but no formal result in the literature as far as I know). I suggest more details about the hashing methods should be included to help the readers better understand the similarity preserving property of LSH and why the method works. More references on LSH methods and applications can also be added to help the readers appreciate the approach.

For this purpose, I think Section 3.2 and 3.3 have something in common and can be combined and compressed a little bit. This could free up some space for more introduction/related context on the theoretical foundation of LSH/SimHash and VSRP.

2. There might be other LSH methods used in HASTE, but they are not tested in the paper. I'm curious about the impact of $s$ (which corresponds to different types of projection) empirically. In the experiments, you just fixed $s$. An ablation study on $s$ could be useful and make the paper more self-contained. What's the result for $s=0$? What about using Gaussian projection instead of $\pm 1$ when $s=0$? etc.

Also, for the cosine similarity, another possible approach might be the count-sketch (CS) type algorithm and some later developments:

[Moses Charikar, Kevin Chen, and Martin Farach-Colton. Finding frequent items in data streams, Theoretical Computer Science, 2004]

[Kilian Q. Weinberger et al. Feature hashing for large scale multitask learning, ICML 2009]

[Ping Li and Xiaoyun Li. OPORP: One Permutation + One Random Projection, KDD 2023]

A signed (1-bit) alternative of CS is SignOPORP in

[Ping Li and Xiaoyun Li. Differential Privacy with Random Projections and Sign Random Projections, Arxiv 2023]

These algorithms might be even more efficient than VSRP. They split a data vector into $M$ bins and conduct projection within each bin, so they essentially only need one random projection in the most efficient case---If the dimensionality of the data, $d$, is sufficiently large, setting $M = L$ (the number of hash codes) suffices and the complexity is simply $d$. As a compariton, for sparse RP used in HASTE, this complexity is $dL(1-s)$. If $d$ is small, we can repeat CS for several times. For example, when $L=8$, we can set $M=4$ and repeat CS for 2 times.

In all, my point is that, there are other LSH methods that can be used in HASTE, but the paper simply used one of them without mentioning others. It will be better if the authors can more fully explore the possible options empirically. Or, at least, more related methods and approaches should be mentioned in the paper to give the readers a more complete picture and possible alternatives.

**Summary:** The paper presents a good combination of deep networks and LSH, with simple and intuitive idea, and satisfactory empirical performance. The organization and presentation can be improved to include more background and discussion on the technical side (the properties, more related LSH methods, possible alternatives/improvements). I think the paper meets the bar of ICLR.

**Questions:**

Can you provide the distribution of the average similarity between the input vectors $x$ (flattened channels)? This could be an useful and interesting statistics to better motivate the proposed method, and also useful for algorithm design. Perhaps consider also adding it to the paper.

---

> ### Author Response · Authors · 2023-11-17
> **Response to concerns and questions**
>
> We thank the reviewer for taking the time to provide extensive and detailed feedback on our work. We are encouraged that the reviewer perceives our idea as simple and intuitive and the corresponding results as convincing. We are also glad that the paper is found to be easy to follow with clear writing. We address the reviewer's concerns and answer open questions below.
> ## Concerns
> **The locality-sensitive hashing (LSH) variant used in the paper lacks rigorous theory.** We agree that the very sparse random projections used in our proposed HASTE module do not offer the same theoretical foundations as dense Gaussian projections. We intend to follow this avenue of research in future work.
> The choice of sparse projections is mainly motivated by compute budget limitations during runtime. Sparse projections require less overall floating-point operations and allow us to trade multiplications for cheap additions during the hashing process.
> To the best of our knowledge, there are no theoretical results that explicitly draw a link between the collision probability of two vectors and their cosine similarity in the sparse projection case. However, asymptotically, as the dimensionality of vectors grows to infinity, the collision probability converges to the case of dense Gaussian random projections [1, Lemma 2].
>
> **Sections 3.2. and 3.3. can be compressed to give more space to context on theoretical foundations of LSH.** That is a good proposal, and we agree that a more in-depth background on hashing will be beneficial to the reader. We will rework these sections accordingly.
>
> **Potential alternative methods of LSH are not tested in the paper. Algorithms like count-sketches could be even more efficient. A more detailed exploration or discussion of related methods would benefit the paper.** It is correct that we have not explored the usage of count-sketches to replace the hashing component in the HASTE module. We took a general look at count sketches and tensor sketches, but had concerns regarding online runtime. We will provide further motivation for our specific LSH choice and a discussion of related methods such as count-sketches in the revised paper.
>
> ## Questions
> **Can you include an ablation study on $s$, the hyperplane support vector sparsity?** Yes, we believe the addition of such an ablation study will give us insights on the hyperparameter sensitivity and enable broader applicability of our method. While the choice of $s = 2/3$ is also motivated by [1], we acknowledge the need for a more detailed ablation and will add in in the revised paper.
>
> **What happens when setting $s=0$?** In this case, the hyperplane normal vectors are densely filled with entries from $\{-1, 0, 1\}$, which differs from dense Gaussian projections with standard normally distributed vector entries. We will include this special case in our ablation on the hyperparameter $s$.
>
> **What are the results when using dense Gaussian projections**? In this most basic case of LSH with random projections, we use normal vectors with entries drawn from a standard normal distribution. The computational cost increase compared to sparse projections is two-fold: Firstly, the amount of FLOPs required for hashing is increased by using densely filled vectors instead of sparse ones. Secondly, besides additions, we also need to perform multiplications in this case, which are more expensive to compute on a hardware level. We agree that this is an interesting baseline for the reader and will include it in our ablations.
>
> **Can you provide the distribution of average similarity between input vectors / input channels?** Yes, this is an interesting idea. Such a visualization could provide insights into which channels are found to be most similar across varying input data, hinting to little class-specific information. We will add the corresponding visualizations to our paper.
>
>  [1] Ping Li, Trevor Hastie, and Kenneth Church. 2006. Very Sparse Random Projections. In Proceedings of the 12th ACM SIGKDD international conference on Knowledge discovery and data mining (KDD '06). doi: 10.1145/1150402.1150436

---

> > ### Comment · Reviewer_fMDs · 2023-11-23
> >
> > Thanks for the reply. Adding more details on LSH would improve the technical quality of the paper, and it would be better if more hashing approaches like count-sketch and OPORP is tested. More parameters like $s$ or the projection type can also be compared empirically. Please update the paper in the revision accordingly.
> >
> > I will keep my score.

---

### Official Review · Reviewer_988t · 2023-11-02

**Soundness:** 3 good
**Presentation:** 3 good
**Contribution:** 3 good
**Rating:** 6
**Confidence:** 4

**Summary:**

This paper introduces HASTE, a CNN pruning method that is both training and data-free. HASTE leverages locality-sensitive hashing to identify redundant channels, offering a substantial reduction in computational complexity. Experimental results conducted on various image CNN models highlight HASTE's ability to decrease FLOPs in models while maintaining accuracy with only minimal accuracy loss.

**Strengths:**

1. The concept of using hash collisions to assess the redundancy of feature channels is both logical and interesting. The paper also provides a compelling rationale for the need to develop a training and data-free CNN pruning method.

2. The method is presented in a clear and comprehensive manner, with detailed explanations. The effectiveness of the approach is demonstrated through experiments conducted on multiple CNN models with results obtained from two different datasets.

**Weaknesses:**

The main drawback of this paper lies in the analysis of the experimental results.

1. The paper would greatly benefit from visualizations of feature maps after the hashing process. These visualizations would help readers understand which channels are considered redundant in the context of image classification. In other words, by visualizing the feature maps, it would become apparent whether the pruned model genuinely focuses on channels crucial for semantic concept recognition.

2. The paper's experimental focus is primarily on image classification tasks. However, since the core objective of model pruning is to identify redundant feature channels and use the saved channels for accurate recognition of image concepts, it would be valuable to assess whether HASTE performs effectively in other image-related tasks, such as object detection. I strongly recommend that the authors extend their testing to these areas.

**Questions:**

Additionally, there appears to be some missing information in Figure 3(b). It is unclear how the hyperparameter L impacts the method's performance. The values of L should be provided to clarify this aspect.

---

> ### Author Response · Authors · 2023-11-17
> **Response to concerns and questions**
>
> We thank the reviewer for the effort devoted to providing insightful feedback on our work. We are glad that our approach of using hash collisions for finding redundant features is seen as conclusive. We are also pleased to that the reviewer acknowledges the need for data-free pruning methods of CNN models. We address the reviewer's concerns and answer open questions below.
>
> ## Concerns:
>
> **The paper would benefit from visualizations of the pruned feature maps**. This is a good point, and such visualizations would greatly enhance the understanding of our method. We believe that this will be beneficial to the understanding of our paper and will consequently add visualizations in the revised version of our paper.
>
> **Extend testing to other tasks.** We also believe that additional experiments for the downstream usage of our proposed method could provide valuable insights. However, for our research, we focused on image classification, as is customary in the field. Nonetheless, future work includes the extension of our module to said tasks to fortify our claim.
>
> ## Questions:
>
> **Is there information missing in Figure 3(b)?** We agree that this figure does not ideally convey the influence of hyperparameter $L$, as stated in its caption. Each data point represents an accuracy / FLOPs trade-off for a certain model, where, from left to right, the hyperparameter $L$ is increased. The corresponding data points are taken from Table 4 in Appendix A.3. To avoid a cluttered figure, we decided to not label the exact choice of $L$ for every point and instead provide a textual description in the fourth paragraph on page 7. However, as stated by the reviewer, this does not convey the full information and should be changed. We will improve upon this figure for a revised version in the updated paper to clarify this.

---

### Official Review · Reviewer_dkAQ · 2023-11-02

**Soundness:** 2 fair
**Presentation:** 3 good
**Contribution:** 2 fair
**Rating:** 5
**Confidence:** 4

**Summary:**

The paper under review for ICLR 2024 presents a method for compressing convolutional neural networks (CNNs) without the need for additional computational resources for fine-tuning or retraining. The proposed HASTE module is capable of pruning pre-trained models without requiring access to the original training data, which could significantly enhance the accessibility and usability of existing models, especially on less powerful hardware.

The paper includes a detailed analysis of the method's performance on various datasets, such as CIFAR-10 and ImageNet, showcasing the trade-offs between the level of compression (measured in FLOPs reduction and compression ratio) and the resulting top-1 accuracy of the pruned models. The results are reported with mean and standard deviation values, calculated over multiple trials with different random seeds to ensure reproducibility.

The authors also discuss the ethical implications of their work, acknowledging the potential for both positive impacts, like reduced energy and carbon footprint, and negative applications, such as military use or mass surveillance. They express their intention to distance themselves from any harmful uses of their technology.

To further support reproducibility, the authors plan to release their code upon publication and provide a comprehensive overview of their experimental setup, model design choices, and the computational requirements of their method.

The paper seems to contribute to the field of efficient CNN architecture design, offering a solution that balances performance with computational efficiency, which is crucial for deploying deep learning models in resource-constrained environments.

**Strengths:**

Originality:
The HASTE module for CNN compression introduces a novel approach to model pruning that does not require retraining or fine-tuning, which is a significant deviation from traditional methods that often rely on these computationally intensive processes.
The technique's ability to operate without the original training data is particularly innovative, as it addresses a common limitation in scenarios where data may be proprietary or privacy-sensitive.

Quality:
The empirical analysis provided is thorough, with the authors conducting multiple trials to ensure reproducibility. The inclusion of mean and standard deviation values over these trials adds to the robustness of the reported results.
The method's performance on benchmark datasets like CIFAR-10 and ImageNet is well-documented, providing a solid foundation for the claimed contributions.

Clarity:
The paper is well-structured, with a clear exposition of the methodology and results. The authors have made an effort to articulate the motivations behind their work and the potential applications of their proposed method.
The ethical considerations section, although brief, demonstrates an awareness of the broader implications of their work, which adds depth to the paper.

Significance:
The potential impact of the HASTE module is significant, particularly for deploying deep learning models in resource-constrained environments. This could democratize access to advanced AI capabilities, especially in areas with limited computational resources.
The authors' commitment to releasing their code upon publication is commendable and will likely facilitate further research and application of their work, enhancing the paper's significance within the community.

**Weaknesses:**

Clarity on Novelty and Differentiation:
The paper introduces a method for compressing CNNs, which is a well-explored area of research. To better understand the novelty of the proposed HASTE module, the authors should provide a clearer differentiation between their method and existing approaches. A more detailed comparison with state-of-the-art methods, possibly in a tabular form, could help highlight the unique contributions. References to prior works like [Howard et al., 2017] on MobileNets and [Zhang et al., 2018] on Shufflenet could be used to benchmark and contrast the proposed method.

Broader Impact on Different Architectures:
The paper presents results primarily on standard architectures like ResNet and VGG. It would be beneficial to see the impact of the HASTE module on a wider range of architectures, including more recent ones like EfficientNet [Tan and Le, 2019] or Vision Transformers [Dosovitskiy et al., 2021]. This would help validate the generalizability of the method.

Quantitative Analysis of Computational Savings:
While the paper discusses FLOPs reduction, a more comprehensive analysis of the actual computational savings in real-world scenarios would be valuable. This includes memory footprint reduction, inference time measurements on hardware where such models are likely to be deployed, and energy efficiency evaluations.

Robustness and Error Analysis:
The paper could benefit from a robustness analysis, showing how the pruned models perform under different conditions, such as adversarial attacks or out-of-distribution data. An error analysis could also be insightful, particularly if there are specific classes or types of data where the pruned models underperform.

Hyperparameter Sensitivity:
The method's performance appears to be sensitive to the choice of hyperparameters, such as the number of hyperplanes L. A more detailed exploration of this sensitivity, along with guidelines for hyperparameter selection in different scenarios, would be useful for practitioners.

Limitations and Failure Modes:
A discussion on the limitations and potential failure modes of the proposed method would provide a more balanced view. For instance, under what conditions does the HASTE module fail to compress effectively, and what are the trade-offs involved?

Reproducibility and Accessibility:
While the authors have stated their intention to release their code upon publication, ensuring that the code is well-documented and easy to use will be crucial for reproducibility. Additionally, providing pre-trained models and a user-friendly interface could facilitate broader adoption and experimentation by the community.

**Questions:**

Could you elaborate on the specific aspects of the HASTE module that distinguish it from prior work in CNN compression? Are there particular mechanisms or theoretical underpinnings that are unique to your approach?
Have you tested the HASTE module on a broader range of neural network architectures, particularly more recent ones? If not, could you hypothesize on its performance or provide a rationale for the selection of the tested architectures?
Can you provide a more detailed analysis of the computational savings in terms of memory, power, and inference time, particularly on edge devices where such compression techniques are most valuable?
Would you be able to include a robustness analysis against adversarial examples or performance on out-of-distribution data? An error analysis on the types of errors introduced post-compression would also be insightful.
Could you discuss any identified limitations or failure modes of the HASTE module? Under what conditions does it fail to deliver the expected compression or performance?
Have you considered the impact of dataset bias on the compression method, and how might this affect the generalizability of the pruned models?
How does the compression affect performance on different tasks beyond image classification, such as object detection or segmentation?
Can you provide a more comprehensive ethical analysis of the potential negative uses of your technology and propose strategies to mitigate these risks?

**Details Of Ethics Concerns:**

"Our work aims to enable compression of existing convolutional neural networks without having to spend additional computational resources on fine-tuning or retraining. Furthermore, our proposed HASTE module allows for pruning of pre-trained models without access to the training data. This has the potential to increase accessibility and usability of existing models by compressing them for usage on less powerful hardware, even when the training dataset is not publicly available. Furthermore, our method facilitates the employment of computationally cheaper models, reducing energy and carbon footprint induced by the network’s inference. However, for the same reasons, our method could also be used in undesired applications such as for military purposes or mass surveillance. We distance ourselves from any application with negative societal impact."

This statement acknowledges the dual-use nature of the technology, highlighting its potential benefits in terms of accessibility, cost, and environmental impact, while also recognizing the possibility of misuse in contexts that could have a negative societal impact. The authors explicitly distance themselves from such applications.

Privacy, Security, and Safety: The compression technique could enable the deployment of advanced neural networks in resource-constrained environments, which may include devices with potential privacy implications, such as smartphones and surveillance cameras. There is a risk that such technology could be used to enhance surveillance capabilities in a manner that infringes on individual privacy rights.

Potentially Harmful Insights, Methodologies, and Applications: The authors acknowledge the possibility of their method being used for military purposes or mass surveillance. This dual-use nature of the technology necessitates a thorough review to ensure that guidelines are in place to prevent misuse and to promote applications that have a positive societal impact.

An ethics review by experts in AI ethics and dual-use technology is recommended to address these concerns.
The ethical considerations for the paper under review are addressed in the following statement from the document:

---

> ### Author Response · Authors · 2023-11-17
> **Response to concerns and questions**
>
> We thank the reviewer for the time and effort spent on providing insightful feedback on our work. We are encouraged to find that the reviewer describes our proposed approach as novel and innovative. We are also pleased that the reviewer finds our work to be well-structured, our analysis thorough and the impact of our work significant. We address the reviewer's concerns and answer open questions below.
> ## Concerns:
> **A tabular overview of existing approaches would provide more clarity.** We also believe that a simple overview helps the reader to clearly distinguish our contribution from existing work. For this reason, we included a variety of related approaches, their unique requirements and capabilities for dynamic pruning in Table 1.
>
> **Contrast your proposed method to prior works like MobileNets or ShuffleNets.** While these prior works consider small and efficient convolutional network architectures, our work focusses on the compression of large architectures that typically exhibit many redundant features. This is motivated in the first paragraph of our introduction: Highly scalable architectures like ResNets are particularly desirable for their performance and ability to generalize, but are not directly employable on embedded hardware. Models like MobileNets or ShuffleNets introduce larger inductive biases as a trade-off. Testing on these models is not the scope of this paper, but a promising direction for future work.
>
> **An analysis on a broader range of architectures would be beneficial.** We agree that the extension of our HASTE module to other CNN architectures is an interesting direction. As stated above, we will explore this in future work.
>
> **A more detailed analysis regarding memory footprint and latency would be valuable.** This is correct, and we want to include memory benchmarks in the revised version of our paper. Figure 2 hints at the potential for memory reduction that is possible with our method. Further measurements are difficult, as our method is currently not natively supported in modern deep learning frameworks. However, saving memory accesses is crucial for employment on embedded hardware.
>
> **The paper could benefit from an analysis of the method's robustness and potential errors, as well as limitations and failure modes.** Thank you for your proposal. An in-depth analysis of adversarial and out-of-distribution performance would be interesting, especially in comparison to related approaches. We will explore this in future work.
>
> **Explore hyperparameter sensitivities and give guidelines for hyperparameter selection.** We provide an overview of the influence of hyperparameter $L$ in Figure 3(b). Additionally, we give results for various hyperparameter settings in Table 4 and 5 and therefore provide an outline of reasonable hyperparameter choices. We discuss the theoretically motivated setting for hyperparameter $s$ in Section 4.1. An additional ablation for $s$ is planned for the revised paper.
>
> **The availability of pre-trained models and a user-friendly interface would facilitate reproducibility.** We fully agree. All pre-trained models used in the paper are already publicly available and easily accessible through the PyTorch framework. As our module is a plug-and-play replacement for regular convolutional modules, the adaptation of our method is very simple.
>
> ## Questions:
> **What distinguishes HASTE from prior work in CNN compression?**
> Our approach distinguishes itself from existing work for CNN compression by offering instant data-free and dynamic pruning. To the best of our knowledge, the proposed HASTE module is the first method that instantly reduces model complexity, in particular, floating-point operations (FLOPs)
> - in a data-free manner (no training or fine-tuning)
> - without requiring any pre-processing of the model's weights
> - dynamically (dependent on the module's input) during model inference,
>
> which allows us to adapt the model's FLOPs in real time to the availability of hardware resources.
>
> **Are there unique mechanisms / theoretical underpinnings used?** We uniquely provide an extension of approximate nearest neighbor search using locality-sensitive hashing to compress latent feature maps in CNN architectures.
>
> **Does your method exhibit dataset bias?** No. As our method is data-free and adapts to input feature maps on-the-fly without any previous learned adaptation to the data, we do not incorporate biases into our pruning approach.
>
> **How does the compression affect performance of tasks beyond image classification?** This is an interesting question, and we aim to explore image-related tasks in future work. As semantic segmentation and object detection models often use pre-trained classification backbones, we hypothesize that these methods profit in a similar way.
>
> **Can you provide a more comprehensive ethical analysis?** We share your concerns for unintended usage of this technology and will make this more clear in the final version of the paper.

---

### Author Response · Authors · 2023-11-21
**Rebuttal Revision**

We thank all the reviewers for providing valuable feedback to our work. In the revised version of our paper, we included the following changes:
 - Added an ablation for the hyperparameter $s$ as well as a discussion of hyperparameter settings in Appendix C.2, including the special cases of $s=0$ and using dense Gaussian projections
 - Added benchmarks for memory savings in Appendix B.3
 - Included visualizations displaying the pruned feature maps in Appendix D.2
 - Added visualizations displaying the average similarity of input channels in Appendix D.1
 - Added a discussion of static data-free approaches such as Red++ in Section 2
 - Clarified our contribution and its unique properties in Section 1
- Removed the parameter-free claim from the paper (**Edit**: Adhering to the ICLR 2024 Author Guide, we are not allowed to make changes to the abstract during the discussion stage. Therefore, we will remove the parameter-free claim from the abstract for the camera-ready version. All such claims in the main text were removed nonetheless.)
- Included further background on related methods that employ count sketch-type algorithms or k-means clustering in Section 2
 - Improved Figure 3(b) by including the specific choices of hyperparameter $L$
 - Revised the ethics statement to include a more detailed discussion of potential negative uses
 - Restructured the Appendix for better readability
 - Added a remark about the similarity-preserving quality of LSH in Section 3.1

---

### Meta-Review · Area_Chair_GAYL · 2023-12-05

**Metareview:**

This paper proposes a CNN pruning method,  HASTE, that is both training and data-free. It uses  locality-sensitive hashing to identify redundant channels and removes them which greatly reduces inference computational cost. Experimental results show HASTE decreases FLOPs in models while maintaining accuracy with only minimal accuracy loss.

The main strength of this work is that it is new and effective. The main issues of several  reviewers (>=3) include: 1) Since weights aren't pruned in advance, there are no storage reductions; 2) Since the pruning pattern is dynamic, hardware can't easily optimize for latency (channels are non-contiguous and can't exploit cache locality with LSH); 3) relying solely on image-level recognition may be insufficient. Conducting examinations on more fine-grained tasks, such as object detection, would enhance this work in demonstrating how it achieves channel selection.

Since most reviewers have low intentions to accept this work, we cannot accept it.

**Justification For Why Not Higher Score:**

The main issues of several reviewers include:

1) Since weights aren't pruned in advance, there are no storage reductions;

2) Since the pruning pattern is dynamic, hardware can't easily optimize for latency (channels are non-contiguous and can't exploit cache locality with LSH);

3) relying solely on image-level recognition may be insufficient. Conducting examinations on more fine-grained tasks, such as object detection, would enhance this work in demonstrating how it achieves channel selection.

**Justification For Why Not Lower Score:**

N/A

---

### Decision · Program_Chairs · 2024-01-16

Reject